# Theory of active self-organization of dense nematic structures in the actin cytoskeleton

**Waleed Mirza[1,2†], Marco De Corato[3], Marco Pensalfini[1‡], Guillermo Vilanova[1], Alejandro Torres-Sánchez[1,4,5]\*, Marino Arroyo[1,4,6]\***

[1]Universitat Politécnica de Catalunya BarcelonaTech, Barcelona, Spain; [2]Barcelona Graduate School of Mathematics (BGSMath), Bellaterra, Spain; [3]Aragon Institute of Engineering Research (I3A), Universidad de Zaragoza, Zaragoza, Spain; [4]Institute for Bioengineering of Catalonia (IBEC), The Barcelona Institute of Science and Technology (BIST), Barcelona, Spain; [5]European Molecular Biology Laboratory, Barcelona, Spain; [6]Centre Internacional de Mètodes Numèrics en Enginyeria (CIMNE), Barcelona, Spain

**\*For correspondence:**
alejandro.torressanchez@embl.es (AT-S);
marino.arroyo@upc.edu (MA)

**Present address:** [†]European Molecular Biology Laboratory, Barcelona, Spain; [‡]Queen Mary University of London, London, United Kingdom

**Competing interest:** The authors declare that no competing interests exist.

## eLife Assessment

In this study, the authors offer a theoretical explanation for the emergence of nematic bundles in the actin cortex, carrying implications for the assembly of actomyosin stress fibers. As such, the study is a **valuable** contribution to the field of actomyosin organisation in the actin cortex. The theoretical work is **solid** and provides a rigorous theoretical framework to study active self-organisation in acto-myosin systems, including qualitative comparison with experimental observations.

**Abstract** The actin cytoskeleton is remarkably adaptable and multifunctional. It often organizes into nematic bundles such as contractile rings or stress fibers. However, how a uniform and isotropic actin gel self-organizes into dense nematic bundles is not fully understood. Here, using an active gel model accounting for nematic order and density variations, we identify an active patterning mechanism leading to localized dense nematic structures. Linear stability analysis and nonlinear finite element simulations establish the conditions for nematic bundle self-assembly and how active gel parameters control the architecture, orientation, connectivity, and dynamics of self-organized patterns. Finally, we substantiate with discrete network simulations the main requirements for nematic bundle formation according to our theory, namely increased active tension perpendicular to the nematic direction and generalized active forces conjugate to nematic order. Our work portrays actin gels as reconfigurable active materials with a spontaneous tendency to develop patterns of dense nematic bundles.

## Introduction

Actin networks are remarkably dynamic and versatile and organize in a variety of architectures to accomplish crucial cellular functions (*Banerjee et al., 2020*). For instance, isotropic thin actin gels form the cell cortex, which largely determines cell shape (*Chugh and Paluch, 2018*) and motility in confined nonadherent environments (*Blaser et al., 2006*; *Ruprecht et al., 2015*). Polar structures at the edge of adherent cells, either forming filaments as in filopodia or sheets as in lamellipodia (*Blanchoin et al., 2014*), enable cells to probe their environment and crawl on substrates. Nematic

actin bundles form a variety of contractile structures (*Schwayer et al., 2016*), including subcellular rings during cytokinesis (*Reymann et al., 2016*) and cortical repair (*Mandato and Bement, 2001*), supracellular rings during wound healing (*Martin and Lewis, 1992*) or development (*Krieg et al., 2008*), bundle networks during cellularization (*Dudin et al., 2019*), or stress fibers in adherent cells (*Senger et al., 2019*; *Tojkander et al., 2012*; *Tojkander et al., 2015*). Nematic bundles consist of highly aligned and densely packed actin filaments of mixed polarity connected by a diversity of crosslinkers. In vivo and in vitro observations show the key role of actin nucleators and regulators, of myosin activity, and of crosslinkers in the assembly and maintenance of actin bundles (*Banerjee et al., 2020*; *Chugh and Paluch, 2018*; *Blanchoin et al., 2014*; *Schwayer et al., 2016*; *Tojkander et al., 2012*; *Chrzanowska-Wodnicka and Burridge, 1996*; *Thoresen et al., 2011*; *Strehle et al., 2011*; *Laporte et al., 2012*; *Wirshing and Cram, 2017*; *Lehtimäki et al., 2021*; *Öztürk-Çolak et al., 2016*).

Various studies have emphasized the morphological, dynamical, molecular, and functional specificities of different types of actin bundles such as dorsal, transverse, and ventral stress fibers or contractile rings (*Tojkander et al., 2012*; *Tojkander et al., 2015*; *Hotulainen and Lappalainen, 2006*; *Naumanen et al., 2008*; *Lee et al., 2018*). Here, we ask the question of whether, despite this diversity, the ubiquity of actin bundles in different contexts can be explained by the intrinsic ability of the active actomyosin gel to self-organize into patterns of dense nematic structures. Suggestive of such active self-organization, stress fibers often form dynamic highly organized patterns, e.g., involving families of fibers along orthogonal directions (*Senger et al., 2019*; *Tojkander et al., 2015*; *Wirshing and Cram, 2017*; *Hotulainen and Lappalainen, 2006*; *Tee et al., 2015*; *Yolland et al., 2019*; *Jalal et al., 2019*). Other kinds of actin bundles also develop patterns of remarkable regularity. For instance, parallel arrangements of 2-µm-spaced actin bundles serve as templates for extracellular matrix deposition during butterfly wing morphogenesis and determine their iridescence (*Dinwiddie et al., 2014*). Similarly, the morphogenesis of the striated tracheal cuticle in *Drosophila* is pre-patterned by a parallel arrangement of actin bundles spaced by ~1 µm, spanning from subcellular to supracellular and organ scales (*Öztürk-Çolak et al., 2016*; *Hannezo et al., 2015*). Muscle-like actin bundles form regular parallel patterns spanning organs as in *Caenorhabditis elegans* (*Wirshing and Cram, 2017*) or the entire organism of hydra (*Maroudas-Sacks et al., 2021*). Furthermore, dense nematic bundles have been shown to assemble de novo from the sparse isotropic cortex in a process controlled by myosin activity (*Lehtimäki et al., 2021*) and to form a mechanically integrated network with the cortex (*Vignaud et al., 2021*).

Theoretical models of stress fibers often assume from the outset the existence of dense actin bundles, possibly embedded in an isotropic network (*Riedel et al., 2025*), although previous work has studied the emergence of nematic rings as a result of localized activity gradients (*Salbreux et al., 2009*). In the field of active nematics, many theoretical studies have examined the well-known hydrodynamical bend/splay instability arising in uniform active systems with preexisting long-range orientational order (*Aditi Simha and Ramaswamy, 2002*; *Ramaswamy and Rao, 2007*; *Doostmohammadi et al., 2018*). However, the question of how patterns of orientational order arise from an isotropic and uniform system as a result of activity has received very little attention and has been based so far on models that do not capture fundamental physical features of actomyosin gels as discussed below (*Zumdieck et al., 2005*; *Santhosh et al., 2020*). Here, to understand the mechanisms underlying the self-assembly of dense actin bundles from a low-density isotropic cortex, we develop a theory for the self-organization of dense nematic structures in the actomyosin cytoskeleton based on a nematic active gel theory accounting for density variations and compressibility. In our theory, nematic patterning is driven by activity rather than by a more conventional crowding mechanism. Linear stability analysis and fully nonlinear simulations show that, when coupled to nematodynamics, the well-known patterning mechanism based on self-reinforcing flows (*Hannezo et al., 2015*; *Bois et al., 2011*; *Kumar et al., 2014*; *Callan-Jones and Voituriez, 2013*; *Mietke et al., 2019*; *Barberi and Kruse, 2023*) leads to a rich diversity of patterns combining density and nematic order. The geometry and dynamics of the emergent patterns are very similar to those observed in diverse cellular contexts. Finally, we test key assumptions of our phenomenological theory leading to such self-organization using discrete network simulations.

## Results

### Theoretical model

The actomyosin cytoskeleton can be understood as a compressible, active, and viscous fluid gel with orientational order undergoing turnover (*Salbreux et al., 2012*; *Balasubramaniam et al., 2022*). Symmetry-breaking and pattern formation in actomyosin gels is often mediated by the emergence of an advective instability leading to compressible self-reinforcing flows. According to this mechanism, fluctuations in the density of cytoskeletal active units generate gradients in active stress, driving flows, which in turn advect the active units reinforcing the initial fluctuation (*Hannezo et al., 2015*; *Bois et al., 2011*; *Kumar et al., 2014*; *Callan-Jones and Voituriez, 2013*). This kind of advective instability has been invoked to explain cell polarization and amoeboid motility (*Ruprecht et al., 2015*; *Callan-Jones and Voituriez, 2013*; *Bergert et al., 2015*), the formation of the cytokinetic ring (*Mietke et al., 2019*), the formation of periodic dense actin structures during tracheal morphogenesis in *Drosophila* (*Hannezo et al., 2015*), or the emergence of self-sustained dynamical states in actomyosin gels extracted from cells (*Krishna et al., 2024*; *Malik-Garbi et al., 2019*), and has been recently reproduced to some degree in confined reconstituted gels from purified proteins (*Sciortino et al., 2025*). In all of these examples, the actomyosin gel develops sustained compressible flows converging toward regions of high density. Furthermore, the compressive strain rate induced by these active flows has been shown to drive nematic order (*Reymann et al., 2016*; *Salbreux et al., 2009*). Finally, observations on adherent cells show that active contractility is required for actin bundle formation (*Wirshing and Cram, 2017*; *Lehtimäki et al., 2021*; *Hotulainen and Lappalainen, 2006*). Therefore, a theoretical model to understand the self-organization of dense nematic structures in actomyosin gels should consider a compressible and density-dependent fluid capturing the advective instability mentioned above. Furthermore, this model should acknowledge the active nature of the assembly of dense nematic structures and permit extended isotropic phases commonly observed in the actin cortex, possibly coexisting with dense nematic phases (*Lehtimäki et al., 2021*; *Vignaud et al., 2021*), rather than thermodynamically enforcing high nematic order everywhere except at topological defects (*Doostmohammadi et al., 2018*; *Gennes and Prost, 1993*; *Beris and Edwards, 1994*; *Soares e Silva et al., 2011*).

Previous models for dry and dilute aligning active matter develop density patterns (*Zumdieck et al., 2005*; *Chaté, 2020*; *Putzig et al., 2016*) but fail to capture the hydrodynamic interactions of actomyosin gels, whereas models for active nematic fluids either ignore density (*Santhosh et al., 2020*; *Marenduzzo et al., 2007*; *Giomi, 2015*; *Jülicher et al., 2018*; *Pearce, 2020*; *Metselaar et al., 2019*; *Srivastava et al., 2016*; *Pokawanvit et al., 2022*) or account for the density of active particles suspended in an incompressible flow (*Aditi Simha and Ramaswamy, 2002*; *Ramaswamy and Rao, 2007*; *Hatwalne et al., 2004*; *Giomi et al., 2014*), and therefore cannot describe the advective instability and self-reinforcing flows of actomyosin gels. Previous models describing the emergence of nematic patterns from uniform and isotropic states ignore either hydrodynamics (*Zumdieck et al.,*

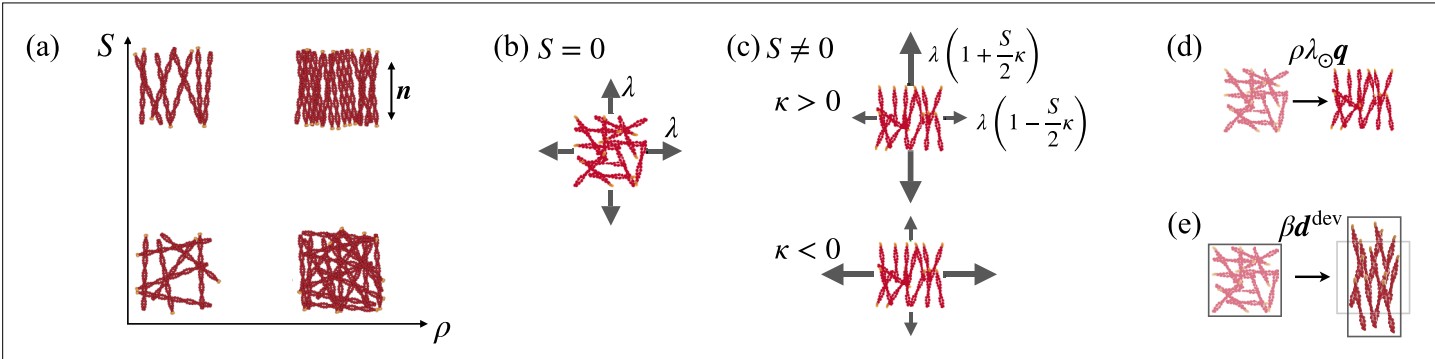

**Figure 1.** Key model ingredients. (**a**) The local state is defined by areal density $\rho$ and by orientational order quantified by the nematic parameter $S$ and by the nematic direction $\boldsymbol{n}$. (**b**) Isotropic active tension $\lambda$ when the network is isotropic ($S = 0$) and (**c**) anisotropic tension when $S \neq 0$, controlled by $\kappa$. Active tension is positive (contractile) in all directions whenever $|\kappa| < 1$, but its deviatoric part is contractile when $\kappa > 0$ and extensile when $\kappa < 0$. Orientational order is driven by (**d**) active forces conjugate to nematic order and characterized by parameter $\lambda_\odot$ and by (**e**) passive flow-induced alignment in the presence of deviatoric rate-of-deformation with coupling parameter $\beta$.

*2005*) or the density and flow compressibility characteristic of actomyosin gels (*Santhosh et al., 2020*), and therefore result in very different instability mechanisms to those presented here. To model a thin layer of actomyosin gel, we summarize next a minimal theory for 2D density-dependent compressible active nematic fluids. In *Mirza et al., 2025*, we provide a systematic derivation of this theory based on a variational formalism of irreversible thermodynamics. This model can be understood as a density-dependent and compressible version of the active nematic theory presented in *Jülicher et al., 2018*, or as a nematic generalization of the isotropic theory used in *Hannezo et al., 2015*. As elaborated in *Mirza et al., 2025*, it is possible to develop an alternative compressible and density-dependent active nematic theory based on the Beris-Edwards formalism (*Santhosh et al., 2020*; *Beris and Edwards, 1994*; *Marenduzzo et al., 2007*; *Giomi, 2015*; *Pearce, 2020*; *Metselaar et al., 2019*; *Giomi et al., 2014*).

In our model, the local state of the system is described by the areal density of cytoskeletal material $\rho(t, \boldsymbol{x})$ and by the network architecture given by the symmetric and traceless nematic tensor $\boldsymbol{q}(t, \boldsymbol{x})$ (see *Figure 1a*), both of which depend on time $t$ and position $\boldsymbol{x}$. A more detailed model could consider separate densities for actin, myosin, and possibly other structural or regulatory proteins. Likewise, in principle, the orientational order of actin and myosin filaments could be described by separate nematic tensors. The nematic tensor can be expressed as $q_{ij} = S\left(n_i n_j - \delta_{ij}/2\right)$, where $\boldsymbol{n}$ is the average molecular alignment, $S = \sqrt{2 q_{ij} q_{ij}}$ the degree of local alignment about $\boldsymbol{n}$, and $\delta_{ij}$ is the identity. We denote by $\boldsymbol{v}(t, \boldsymbol{x})$ the velocity field of the gel. The rate-of-deformation tensor $\boldsymbol{d} = \frac{1}{2}(\nabla \boldsymbol{v} + \nabla \boldsymbol{v}^T)$ measures the local rate of distortion of the fluid, whereas $\boldsymbol{w} = \frac{1}{2}(\nabla \boldsymbol{v} - \nabla \boldsymbol{v}^T)$ measures its local spin, where $\nabla$ is the nabla differential operator. The deviatoric part of the rate-of-deformation tensor is defined by $d_{ij}^{\text{dev}} = d_{ij} - (d_{kk}/2)\delta_{ij}$. The rate of change of $q$ relative to a frame that translates and locally rotates with the flow generated by $\boldsymbol{v}$ is given by the Jaumann derivative $\hat{\boldsymbol{q}} = \partial \boldsymbol{q}/\partial t + \boldsymbol{v} \cdot \nabla \boldsymbol{q} - \boldsymbol{w} \cdot \boldsymbol{q} + \boldsymbol{q} \cdot \boldsymbol{w}$ (*Gennes and Prost, 1993*).

Because we consider a bi-periodic domain $\Omega$, we ignore other boundary conditions. Balance of cytoskeletal mass for a compressible fluid undergoing turnover takes the conventional form (*Hannezo et al., 2015*)

$$\frac{\partial \rho}{\partial t} + \boldsymbol{\nabla} \cdot (\rho \boldsymbol{v}) - D \Delta \rho + k_d (\rho - \rho_0) = 0, \tag{1}$$

where the second term models advection of cytoskeletal material by flow, the third term models diffusion with $D$ being an effective diffusivity, and the last term models cytoskeletal turnover, where $\rho_0$ is the steady-state areal density and $k_d$ is the depolymerization rate. Note that for a uniform, steady-state, and quiescent gel, the first three terms vanish and $\rho(t, \boldsymbol{x}) = \rho_0$.

Force balance in the gel can be expressed as

$$\rho \gamma \boldsymbol{v} = \boldsymbol{\nabla} \cdot \boldsymbol{\sigma}, \tag{2}$$

where $\gamma > 0$ models a viscous drag with the surroundings (e.g. the plasma membrane) and $\boldsymbol{\sigma} = \boldsymbol{\sigma}^{\text{nem}} + \boldsymbol{\sigma}^{\text{diss}} + \boldsymbol{\sigma}^{\text{act}}$ is the stress tensor in the gel, which in 2D has units of tension and which includes a contribution coming from the nematic free energy, a dissipative contribution, and an active contribution.

The nematic stress $\boldsymbol{\sigma}^{\text{nem}}$ follows from a standard derivation adapted here to a density-dependent material. It derives from the free energy $\mathcal{F} = \int_\Omega \rho f(\boldsymbol{q}, \nabla \boldsymbol{q}) \, dS$, where

$$f(\boldsymbol{q}, \nabla \boldsymbol{q}) = \frac{1}{2} a S^2 + \frac{1}{8} b S^4 + \frac{1}{2} L \left| \nabla \boldsymbol{q} \right|^2 \tag{3}$$

is the classical Landau expansion of free-energy density per unit mass, with $a$ and $b > 0$ susceptibility parameters, and $L > 0$ the Frank constant. When $a < 0$, the first term favors equilibrium nematic ordering, e.g., due to crowding in very dense gels of elongated filaments. Otherwise, the susceptibility terms entropically favor isotropic states with small $S$. In the actin cytoskeleton, this can model the random orientation of filaments as a result of the entropic fluctuations of filaments and their nucleators. The last term penalizes sharp gradients in the nematic field, which can result from the bending rigidity of actin filaments. A lengthy but standard calculation leads to the so-called molecular field (*Mirza et al., 2025*)

$$h_{ij} = -\frac{\delta \mathcal{F}}{\delta q_{ij}} = -\rho \left( 2a + bS^2 \right) q_{ij} + L\nabla_k \left( \rho \nabla_k q_{ij} \right), \tag{4}$$

and to the explicit form of the nematic nonsymmetric stress tensor

$$\sigma_{ij}^{\text{nem}} = -\rho \frac{\partial f}{\partial \nabla_j q_{lk}} \nabla_i q_{lk} + q_{ik} h_{jk} - q_{jk} h_{ik} = L \left[ -\rho \nabla_i q_{kl} \nabla_j q_{kl} + q_{ik} \nabla_l \left( \rho \nabla_l q_{jk} \right) - q_{jk} \nabla_l \left( \rho \nabla_l q_{ik} \right) \right]. \tag{5}$$

The dissipative part of the stress

$$\sigma_{ij}^{\text{diss}} = \rho \left[ 2\eta \left( d_{ij} + d_{kk}\delta_{ij} \right) + \beta \hat{q}_{ij} \right], \tag{6}$$

includes a viscous stress controlled by the gel viscosity parameter $\eta$ (**Salbreux et al., 2009**) and a stress resulting from changes in the nematic field controlled by the coupling parameter $\beta < 0$ (**Reymann et al., 2016**). The term involving $\beta$ can be understood as a stress in the gel arising from the drag induced by filaments as they reorient relative to the underlying hydrodynamic flow. Finally, we assume that the active stress resulting from the mechanical transduction of chemical power has an isotropic component and an anisotropic component oriented along the nematic tensor following

$$\sigma_{ij}^{\text{act}} = \rho \left( \lambda \delta_{ij} + \lambda_{\text{aniso}} q_{ij} \right) = \rho\lambda \left( \delta_{ij} + \kappa q_{ij} \right). \tag{7}$$

The activity parameter $\lambda$ controls the isotropic tension and is contractile for $\lambda > 0$, as assumed here. The additional activity parameter $\lambda_{\text{aniso}}$, or equivalently $\kappa = \lambda_{\text{aniso}}/\lambda$, controls the deviatoric (traceless) component of active tension. This component is anisotropic and can be positive or negative parallel or perpendicular to the nematic direction depending on the sign of $\kappa$. When $|\kappa| < 1$, then the total active tension remains positive in all directions, with a larger magnitude parallel to the nematic direction when $\kappa > 0$ (contractile deviatoric component) and perpendicular to it when $\kappa < 0$ (extensile deviatoric component). We note that the isotropic component of active tension is meaningful here because our active gel is compressible. When order is low ($S \approx 0$), active tension is isotropic (**Figure 1b**), whereas when order is high, active tension becomes anisotropic (**Figure 1c**). We can interpret that active tension along the nematic direction reflects the sliding of antiparallel fibers driven by myosin motors, and whereas active tension perpendicular to it reflects the out-of-equilibrium binding of bundling proteins or myosins (**Blanchoin et al., 2014**; **Harris et al., 2006**; **Courson and Rock, 2010**; **Schuppler et al., 2016**; **Li et al., 2017**; **Nandi, 2018**; **Ennomani et al., 2016**; **Chen et al., 2020**).

Balance of the generalized forces power conjugate to $\hat{q}$ also includes viscous, elastic-nematic, and active contributions and takes the form

$$\eta_{\text{rot}}\hat{q} + \beta d^{\text{dev}} - \frac{1}{\rho}h - \rho\lambda_{\odot}q = 0. \tag{8}$$

In this expression, $\eta_{\text{rot}}$ is a nematic viscous coefficient. The second term models alignment induced by the rate of deformation of the flow, e.g., with compression/extension driving alignment perpendicular/parallel to the velocity gradient (**Figure 1e**), as experimentally observed in **Reymann et al., 2016**. This term involves the same coefficient $\beta$ as the last term in **Equation 6** because of Onsager's reciprocity relations, and the entropy production inequality requires that $2\eta\eta_{\text{rot}} - \beta^2 \geq 0$ (**Mirza et al., 2025**). The third term is a thermodynamic force driven by $\mathcal{F}$. In agreement with the observations that nematic ordering in actomyosin gels is actively driven, we assume $a > 0$, and therefore **Equation 4**, **Equation 8** show that this term tends to restore isotropy. The last term is an active generalized force controlled by the activity parameter $\lambda_{\odot} \geq 0$ tending to further align filaments (**Figure 1d**; **Reymann et al., 2016**). This term is linear in $\rho$ because in the expansion $\bar{\lambda}_{\odot} + \rho\lambda_{\odot}$, the constant contribution $\bar{\lambda}_{\odot}$ can be subsumed by the susceptibility parameter $a$ (**Salbreux et al., 2009**). Thus, the active term acts as a negative density-dependent susceptibility. When $c_0 = 2a - \rho_0\lambda_{\odot} < 0$, the system can sustain a uniform quiescent state with $\rho(x, t) = \rho_0$, $v(x, t) = 0$ and a nonzero nematic order parameter $S_0^2 = -c_0/b$. Even if $c_0 > 0$, and hence the uniform quiescent state is devoid of order, pattern formation can induce density variations such that $2a - \rho\lambda_{\odot}$ becomes negative locally and actively favors local nematic order. Physically, the term $-\rho\lambda_{\odot}q$ implements the notion that the binding of a bundling protein, which drives active alignment, is more probable when two filaments are in close proximity and nearly aligned, a situation favored by high density and nematic order.

The nonlinearity of the coupled system of partial differential equations given by the balance laws in *Equation 1*, *Equation 7*, *Equation 8*, along with the constitutive relations in *Equation 4*, *Equation 5*, *Equation 6*, *Equation 7*, has different sources summarized below. The theory presents nonlinearities intrinsic to transport equations in the advective term of *Equation 1* and in the definition of the Jaumann derivative of the nematic tensor. Furthermore, nonlinearities in *q* in the constitutive relations result from the standard nematic free energy adopted here. Our hypothesis that material properties in the gel are proportional to density and our thermodynamically consistent derivation of the theory (*Mirza et al., 2025*) result in further nonlinearities involving density in the constitutive relations. Finally, the nonlinearity involving density and the nematic field in the last term of *Equation 8* has been discussed in the previous paragraph.

Our theory has three active parameters, $\lambda$, $\kappa$ and $\lambda_{\odot}$, all reflecting the conversion of chemical power into mechanical power in the network. The magnitude and the modes of chemomechanical transduction should depend on the molecular architecture of the network (*Chugh, 2017*; *Koenderink and Paluch, 2018*), e.g., the stoichiometry of filaments, crosslinkers, and myosins, or the length distribution of filaments. Accordingly, we allow these parameters to vary independently.

By freezing an isotropic state, $S=0$, our model reduces to an orientation-independent active gel model, which develops periodic patterns driven by self-reinforcing active flows sustained by diffusion and turnover (*Hannezo et al., 2015*). In the present model, however, translational, orientational, and density dynamics are intimately coupled through the terms involving $\beta$, $\lambda_{\odot}$, $\kappa$ and $L$.

We readily identify the hydrodynamic length $\ell_s = \sqrt{\eta/\gamma}$, above which friction dominates over viscosity, the Damköhler length $\ell_D = \sqrt{D/k_d}$ above which reactions dominate over diffusion, and the nematic length $\ell_q = \sqrt{L/|2a - \lambda_{\odot}\rho_0|}$. Nondimensional analysis reveals a set of nondimensional groups that control the system behavior, namely the nondimensional turnover rate $\bar{k}_d = \ell_s^2/\ell_D^2$, the Frank constant $\bar{L} = L/(\eta D)$, the susceptibility parameters $\bar{a} = a/(\gamma D)$ and $\bar{b} = b/(\gamma D)$, the drag coefficients $\bar{\eta}_{\text{rot}} = \eta_{\text{rot}}/\eta$ and $\bar{\beta} = \beta/\eta$, the nematic activity coefficient $\bar{\lambda}_{\odot} = \rho_0\lambda_{\odot}/(\gamma D)$, and the active tension parameters $\bar{\lambda} = \lambda/(\gamma D)$ and $\kappa$ (Appendix B). The full list of material parameters for each figure is given in *Appendix 4—table 1* and *Appendix 4—table 2* and justified in Appendix D .

## Onset and nature of pattern formation

To examine the role of nematic order in the emergence of various actin architectures, we performed linear stability analysis of our model particularized to 1D, whose dynamical variables are velocity, density, and nematic order, $v(x,t)$, $\rho(x,t)$, and $q(x,t)$, along $x$, where $q > 0\ (<0)$ corresponds to a nematic orientation **n** parallel (perpendicular) to the *x*-axis (Appendix A). We first focused on the case $c_0 = 2a - \rho_0\lambda_{\odot} > 0$ to examine the loss of stability of a uniform, isotropic, and quiescent steady state ($\rho(x,t) = \rho_0$, $v(x,t) = 0$, $q(x,t) = 0$) by increasing the master activity parameter $\lambda$ and identifying the most unstable modes. This allowed us to determine a threshold activity for pattern formation and the wavelength of the emerging pattern (Appendix C). Since the exact evaluation of such quantities requires solving nonlinear equations, we derived explicit expansions in the limit of small $L$ for the critical contractile activity

$$\lambda_{\text{crit}} \approx \lambda_{\text{crit},0}\left[1 - \frac{1}{2}\frac{\ell_s}{\ell_D}\left(1 + 2\frac{\ell_s}{\ell_D}\right)\delta\right] + \mathcal{O}(\delta^2),\tag{9}$$

where $\lambda_{\text{crit},0} = \left(\sqrt{\gamma D} + 2\sqrt{k_d\eta}\right)^2 = \gamma D(1 + 2\sqrt{\ell_s/\ell_D})^2$ and $\delta = \gamma D\kappa\beta/(2\eta c_0)$, and for the corresponding wavenumber

$$\nu_{\text{crit}}^2 \approx \nu_{\text{crit},0}^2\left[1 + \frac{1}{8}\left(1 + 2\frac{\ell_s}{\ell_D}\right)^2\delta\right] + \mathcal{O}(\delta^2),\tag{10}$$

where $\nu_{\text{crit},0}^2 = \left[k_d\gamma/(4\eta D)\right]^{1/2} = 1/(2\ell_s\ell_D)$.

When $\kappa = 0$ or $\beta = 0$, and hence $\delta = 0$, we recover the predictions of an active gel model not accounting for network architecture (*Hannezo et al., 2015*), $\lambda_{\text{crit}} = \lambda_{\text{crit},0}$ and $\nu_{\text{crit}} = \nu_{\text{crit},0}$. The expression for $\lambda_{\text{crit},0}$ shows that the instability takes place when activity is large enough to overcome the effect of diffusion and turnover, tending to uniformize density, and that of friction and viscosity, tending to suppress flows. Active tension anisotropy ($\kappa \neq 0$) and flow-induced alignment ($\beta < 0$) couple the

instability of *Hannezo et al., 2015*, to nematic order, changing the nature of pattern formation (see *Equation 9*, *Equation 10*) and the expression for $\delta$. This leads to quantitative changes in critical tension and wavenumber, which can be very significant depending on the ratio of hydrodynamic and Damköhler lengths and on the strength and sign of nematic coupling. The nematic corrections increase as $c_0$, close to the point where the uniform quiescent state develops spontaneous order. We thus studied separately the regime $0 < c_0 \ll 1$, finding analogous expansions for the critical tension and wavenumber in terms of $\delta = \gamma D \kappa \beta / (2\eta L)$. Interestingly, *Equation 9* shows that the activity threshold is reduced, and hence pattern formation is facilitated, when $\kappa < 0$, i.e., when active tension is larger perpendicular to the nematic direction. Besides modifying critical tension and wavenumber, the present model predicts that the dynamical modes with self-reinforcing flows generate patterns where high density co-localizes with high nematic order.

To test the validity of this analysis and further understand the system beyond the onset of pattern formation, we performed fully nonlinear finite element simulations in a periodic square 2D domain (*Mirza et al., 2025*; *Mirza, 2025*), with a domain size chosen to be $8\,\ell_{\text{pattern}}$, where $\ell_{\text{pattern}} = 2\pi/\nu_{\text{crit}}$ is the pattern lengthscale predicted by linear stability analysis. In these simulations, we increased the activity parameter $\lambda$ beyond the instability starting from a quiescent uniform state. We found that

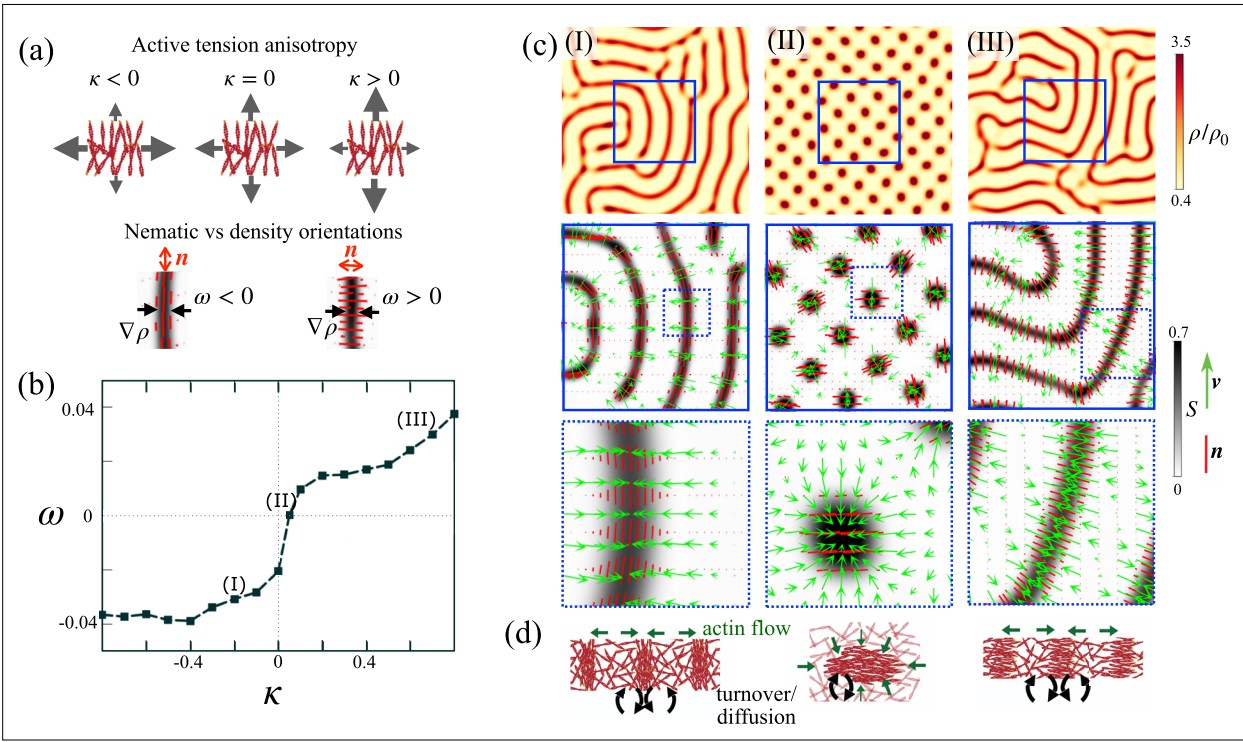

**Figure 2.** Active patterns coupling nematic order and density driven by self-reinforcing flows. (**a**) Illustration of the dimensionless parameters characterizing active tension anisotropy ($\kappa$) and pattern architecture, quantified by the relative orientation of nematic order and high-density structures ($\omega = \ell_s^2/(\rho_0^2|A|)\int_A \nabla\rho \cdot \boldsymbol{q} \cdot \nabla\rho \, dS$). (**b**) Order parameter of pattern architecture $\omega$ as a function of active tension anisotropy $\kappa$ obtained from nonlinear simulations, showing transition from states with nematic direction parallel to high-density structures ($\omega < 0$, fibrillar patterns) for $\kappa < 0$ to states with nematic direction perpendicular to high-density structures ($\omega > 0$, banded patterns with perpendicular nematic organization) for $\kappa > 0$. Because $|\kappa| < 1$, the active tension is always positive in all directions. (**c**) Map of density, nematic order $S$, nematic direction (red segments), and flow field (green arrows) for quasi-steady fibrillar (I) and banded (III) patterns, and for a transition pattern of high-density droplets with high nematic order (II) corresponding to nearly isotropic active tension. (**d**) Illustration of the out-of-equilibrium quasi-steady states, maintained by self-reinforcing flows, diffusion, and turnover.

The online version of this article includes the following video and figure supplement(s) for figure 2:

**Figure supplement 1.** Lengthscale of the pattern.

**Figure 2—video 1.** Pattern formation in an active gel model not accounting for nematic order.

https://elifesciences.org/articles/93097/figures#fig2video1

**Figure 2—video 2.** Pattern formation for different values of the tension anisotropy coefficient $\kappa$, leading to (I) fibrillar patterns for $\kappa < 0$, (II) tactoids for $\kappa \approx 0$, and (III) banded patterns for $\kappa > 0$.

https://elifesciences.org/articles/93097/figures#fig2video2

the linear stability analysis very accurately predicts the activity thresholds within 2% across a wide range of parameters. Furthermore, we found that the linear estimate for the wavenumber in *Equation 10* characterizes well the nonlinear patterns, as quantified in *Figure 2—figure supplement 1* and illustrated visually by the density patterns in *Figure 2* showing the entire $8\,\ell_{\text{pattern}} \times 8\,\ell_{\text{pattern}}$ computational domain. In the nonlinear regime, the exponentially growing instabilities eventually reach out-of-equilibrium quasi-steady-state patterns involving self-reinforcing flows toward regularly spaced regions of high density surrounded by a low-density matrix, as previously reported for various compressible and density-dependent active gel models (*Hannezo et al., 2015*; *Bois et al., 2011*; *Kumar et al., 2014*; *Callan-Jones and Voituriez, 2013*; *Mietke et al., 2019*; *Barberi and Kruse, 2023*). In the absence of nematic coupling ($\beta = \kappa = \lambda_{\odot} = 0$), these high-density domains are droplets arranged in a regular hexagonal lattice with $S = 0$ throughout the domain as previously reported (*Kumar et al., 2014*, *Figure 2—video 1*). In contrast, for a generic parameter set with finite $\beta$, $\kappa$, and $\lambda_{\odot}$, high-density domains adopt elongated configurations or bands where order is high, surrounded by a low-density and low-order matrix (*Figure 2b*, *Figure 2—video 2*). Thus, our nematic active gel develops out-of-equilibrium, localized nematic states starting from an isotropic state through a mechanochemical mechanism, which unlike those in *Zumdieck et al., 2005*; *Santhosh et al., 2020*, exhibit localization of both density and nematic order, and hence resemble dense nematic structures embedded in a low-density isotropic actin cortex. Furthermore, the shape and internal architecture of dense and nematic phases are qualitatively modified by the nematic coupling.

Our simulations show that self-reinforcing flows develop along the direction of largest active tension, and consequently, the pattern architecture depends on the sign of $\kappa$ (*Figure 2c*). For $\kappa < 0$, the system self-organizes into high-density and high-order bands, where nematic direction is parallel to their axis, in what we call *fibrillar pattern* (*Figure 2bI*). Instead, for $\kappa > 0$, nematic order is perpendicular to the axis of the bands, in what we call banded pattern (*Figure 2bIII*). To systematically study the effect of active tension anisotropy, we varied $\kappa$ between –0.8 and 0.8 while keeping all other nondimensional groups fixed and setting $\lambda$ to be 1.3 times the critical activity. We defined the order parameter $\omega$ (*Figure 2a*), allowing us to distinguish between banded ($\omega > 0$) and fibrillar ($\omega < 0$) organizations. We found a sharp transition between fibrillar and banded regimes around $\kappa \approx 0$, during which elongated high-density and high-order domains fragment into nematic droplets or tactoids (*Weirich et al., 2017*; *Weirich et al., 2019*; *Figure 2bII*).

Our results for $\kappa < 0$, leading to self-organized dense nematic fibrillar patterns from an isotropic low-density network, are in agreement with evidence suggesting that stress fibers can assemble from the actin cortex without the involvement of stress fiber precursors or actin polymerization at focal adhesions (*Lehtimäki et al., 2021*). They also agree with observations showing that actin bundles form a mechanical continuum with the surrounding sparse and isotropic cortex (*Vignaud et al., 2021*). Their morphology and patterning dynamics is strikingly reminiscent of actin microridges, formed at the apical surfaces of mucosal epithelial cells (*Depasquale, 2018*; *van Loon et al., 2020*). As discussed earlier, the condition $\kappa < 0$ implies that the deviatoric component of active tension is extensile. In agreement with widely studied incompressible extensile nematic systems, here, the material is drawn perpendicular to the nematic direction, but rather than being expelled along the nematic direction as required from incompressibility (*Doostmohammadi et al., 2018*), here, it is recycled by disassembly and turnover (*Figure 2c, d*, left, and *Equation 1*). Finally, we note the similarity in terms of density and nematic architecture between our fibrillar patterns and those emerging in other active systems through different mechanisms of self-organization, including polar motile filaments (*Huber et al., 2018*; *Denk and Frey, 2020*) or mean-field models of dry mixtures of microtubules and motors (*Maryshev et al., 2019*).

## Requirements for fibrillar and banded patterns

At linear order, our theory shows that the distinctly nematic self-organization requires both flow-induced alignment ($\beta$) and active tension anisotropy ($\kappa$), whereas no condition is required on nematic activity ($\lambda_{\odot}$). We performed further simulations to establish the requirements for fibrillar and banded active patterning in the nonlinear regime. We found that both banded and fibrillar patterns readily form for $\beta = 0$ and finite $\kappa$, yet a finite value of $\beta$ enhances fibrillar formation, leading to longer and more stable dense bands and hindering banded organization (*Figure 3—video 1*). This behavior is

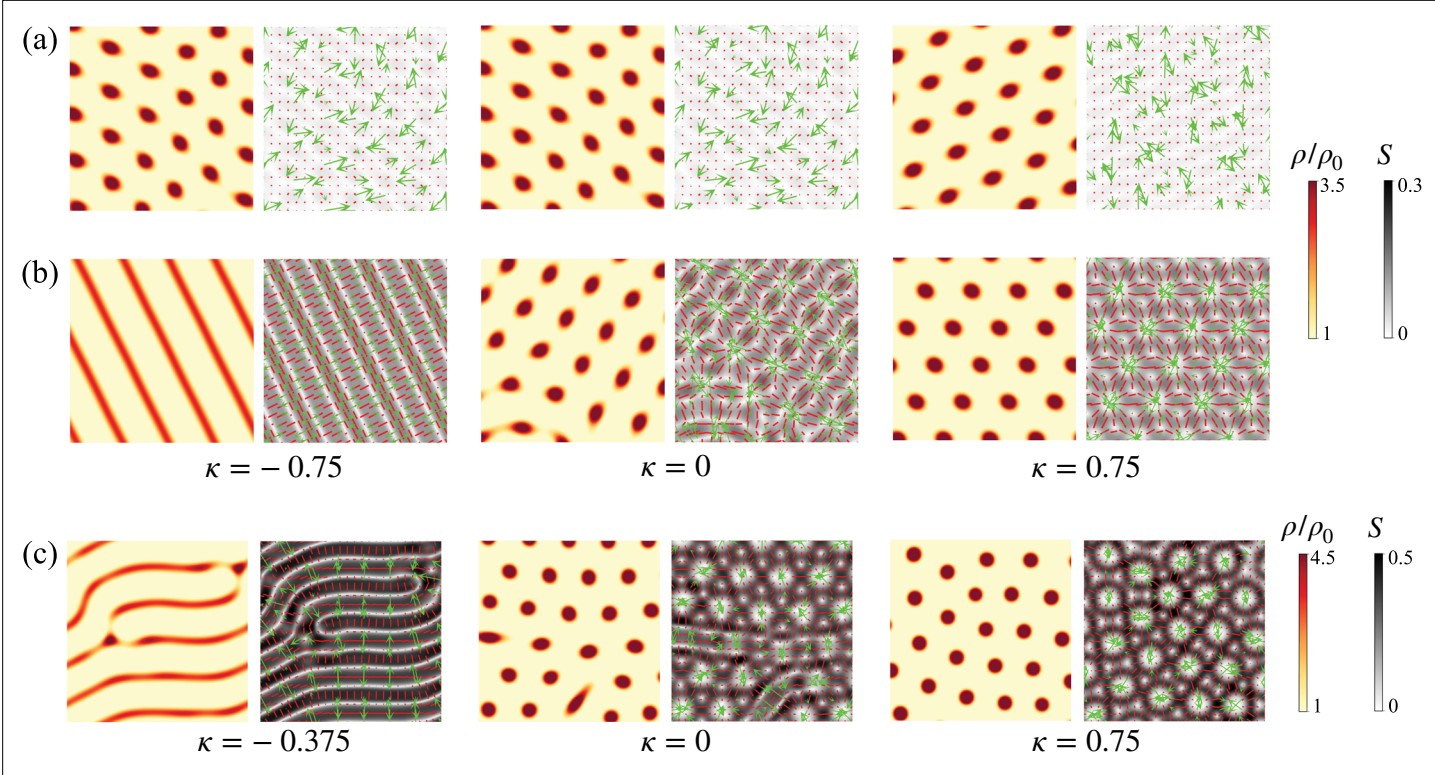

**Figure 3.** Pattern formation for a range of values of anisotropic active parameter $\kappa$ in the limit $\lambda_\odot \to 0$. (**a**) Pattern formation for material parameters used in **Figure 2** except for $\lambda_\odot = 0$ while leaving $c_0$ unchanged. (**b**) Here, in addition to $\lambda_\odot = 0$, we set $\beta^2 = 2\eta\eta_{\mathrm{rot}}$ to the largest value allowed by the entropy production inequality. (**c**) Same parameters as in (**b**), except for an increase in friction as detailed in **Appendix 4—table 1**.

The online version of this article includes the following video for figure 3:

**Figure 3—video 1.** Effect of flow-alignment coupling coefficient $\beta$ on pattern formation for positive and negative tension anisotropy coefficient $\kappa$.

https://elifesciences.org/articles/93097/figures#fig3video1

expected since the velocity gradients of the self-reinforcing flows tend to align filaments parallel to high-density bands due to $\beta d^{\mathrm{dev}}$ in **Equation 8**.

The active nematic coefficient $\lambda_\odot$ modifies the onset of pattern formation through $c_0$ according to linear stability analysis. Apart from this linear effect, it should also contribute to the condensation of nematic order in high-density regions since it appears multiplied by $\rho$ in **Equation 8**. To examine this nonlinear effect, we perform simulations with $\lambda_\odot = 0$ but keeping $c_0$ fixed. This leads to very different patterns without clear elongated structures and very mild nematic patterning (**Figure 3a**). Enhancing nematic patterning by considering the largest thermodynamically allowed value of $|\beta|$ leads to elongated structures for $\kappa < 0$, but rather than high-order co-localizing with high density, the nematic field develops domains with 90° angles between high- and low-density regions (**Figure 3b**), as observed in other active nematic systems (**Zumdieck et al., 2005**). This architecture is enhanced by higher friction $\gamma$ (**Figure 3c**). Thus, the architectures found for $\lambda_\odot = 0$ and $\kappa < 0$ are distinct from the fibrillar pattern described previously. Similarly, rather than bands with order perpendicular to their axis, for $\lambda_\odot = 0$, $\kappa > 0$, and high $|\beta|$, we found patterns of nematic asters (high-density droplets with radial nematic organization around them) (**Figure 3b, c**).

Together, these results show that active tension anisotropy ($\kappa \neq 0$) and nematic activity ($\lambda_\odot \neq 0$) are necessary and sufficient for nematic self-organization into fibrillar or banded patterns, with flow-induced alignment ($\beta < 0$) favoring fibrillar organization. Furthermore, they show that while **Equations 9 and 10** accurately predict the activity threshold and the wavenumber of the nonlinear patterns, the linear stability analysis does not capture the architecture of the patterns, nor the key role of $\lambda_\odot$, in the nonlinear regime.

## Morphological and dynamical diversity of self-organized fibrillar patterns

Given the morphological and dynamical diversity of nematic bundles in actin gels across cell types, geometric confinement, mechanical environment, or biological and pharmacological treatments (*Dudin et al., 2019*; *Yolland et al., 2019*; *Jalal et al., 2019*; *Alert et al., 2022*; *Gupta et al., 2015*; *Xia et al., 2019*), we varied model parameters to examine the architectures predicted by our active gel model, focusing on $\kappa < 0$. Significant changes in the effective parameters of our active gel model are reasonable since the active mechanical properties of actomyosin gels strongly depend on micro-architecture both in reconstituted systems and in cells (*Ennomani et al., 2016*; *Chugh, 2017*).

With our default parameter set, bundle junctions and free ends are unfavorable and reorganize during pattern formation to annihilate as much as possible (*Figure 4—video 1I*). However, because the initial state of the system is isotropic but fibrillar patterns are not, this coarsening process leads to frustrated labyrinth patterns with domains and defects, which, depending on parameters, can remain frozen in quasi-steady states as in *Figure 4aI*. We then asked the question of whether an orientational bias, which physically may be caused by cytoskeletal flows, boundaries, or directed polymerization (*Hotulainen and Lappalainen, 2006*), could direct the pattern and result in fewer defects. We first slightly modified the system by including a small anisotropic strain rate, according to which friction is computed relative to an elongating background. This directional bias is sufficient to produce well-oriented defect-free patterns aligned with the direction of elongation (*Figure 4aII*, *Figure 4—video 1II*). Alternatively, we considered $c_0$ to be slightly negative, leading to the uniform and nematic steady state $\rho(x,t) = \rho_0$, $v(x,t) = 0$, and $q(x,t) = q_0 = \pm\frac{1}{2}\sqrt{-c_0/b}$. The linear stability analysis around this state and further nonlinear simulations show that the essential phenomenology of *Equation 9*, *Equation 10*, and *Figure 2* is not altered by the slight preexisting order (Appendix C). Again, the weak preexisting order provides sufficient bias to direct pattern orientation (*Figure 4aIII*). Thus, our model indicates that an anisotropic bias can guide and anneal nematic fibrillar patterns, in agreement with remark-ably oriented patterns of actin bundles in elongated cells (*Dinwiddie et al., 2014*), as a result of uniaxial cellular stretch (*Wirshing and Cram, 2017*; *van Loon et al., 2020*), or on anisotropically curved surfaces (*Öztürk-Çolak et al., 2016*; *Hannezo et al., 2015*).

At longer times, the nematic bundles in *Figure 4aIII* develop secondary active instabilities leading to coordinated bending, curvature amplification, defect nucleation, and annihilation (*Figure 4—video 1III*) in a behavior reminiscent of active extensile systems (*Aditi Simha and Ramaswamy, 2002*; *Ramaswamy and Rao, 2007*; *Doostmohammadi et al., 2018*). This possibility is far from obvious because, even though the deviatoric part of active tension is extensile since $\kappa < 0$, total active tension is contractile since $|\kappa| < 1$ (*Figure 1c*). To systematically examine this point, we performed simulations at higher active tension anisotropies $\kappa = -0.8$, which we compared with our reference $\kappa = -0.2$ (*Figure 4b*). While in our reference system, bundles behave like contractile objects tending to straighten, for $\kappa = -0.8$ they behave like extensile objects enhancing curvature (see blue insets), which results in continuous defect nucleation as highly bent bundles destabilize and fragment, as well as defect annihilation as pairs of free ends merge to reorganize the network (see second inset in *Figure 4bII*, in a behavior akin to active turbulence *Verkhovsky et al., 1997*). See *Figure 4—video 2* for an illustration and for the slightly extensile case $\kappa = -0.5$. To further substantiate this contractile vs extensile behavior of the nematic structures emerging from a contractile active gel with exten-sile deviatoric behavior, we plotted the difference between the stress along the nematic direction and perpendicular to it, $\sigma_\parallel - \sigma_\perp$. We note that for $\kappa < 0$, as required to obtain fibrillar patterns, the active component of this quantity is negative, $\sigma_\parallel^{act} - \sigma_\perp^{act} < 0$ (*Figure 1c*). In the case of bundles with contractile phenomenology, i.e., a tendency to straighten *Figure 4bI*, we found that $\sigma_\parallel - \sigma_\perp$ was posi-tive on the dense bundles and nearly zero in between, indicating that bundles are contractile struc-tures embedded in a largely isotropic matrix. Conversely, in bundles with extensile phenomenology *Figure 4bII*, we found intricate stress patterns with regions of negative $\sigma_\parallel - \sigma_\perp$. The extensile tension pattern for $\kappa = -0.8$ is more clearly appreciated in 1D simulations (*Figure 4—figure supplement 1*), where the dense nematic structures cannot fragment as a result of the bend-type instability. This figure also shows how the competition between active and viscous stresses dictates the emergent contrac-tile vs extensile behavior of the nematic bundles. In the contractile case ($\kappa = -0.2$), even if $\sigma_\parallel^{act} < \sigma_\perp^{act}$, the negative viscous stresses due to the self-reinforcing flows are significantly larger in magnitude perpendicular to the nematic direction, resulting in $\sigma_\parallel > \sigma_\perp$. In summary, our theory predicts that a

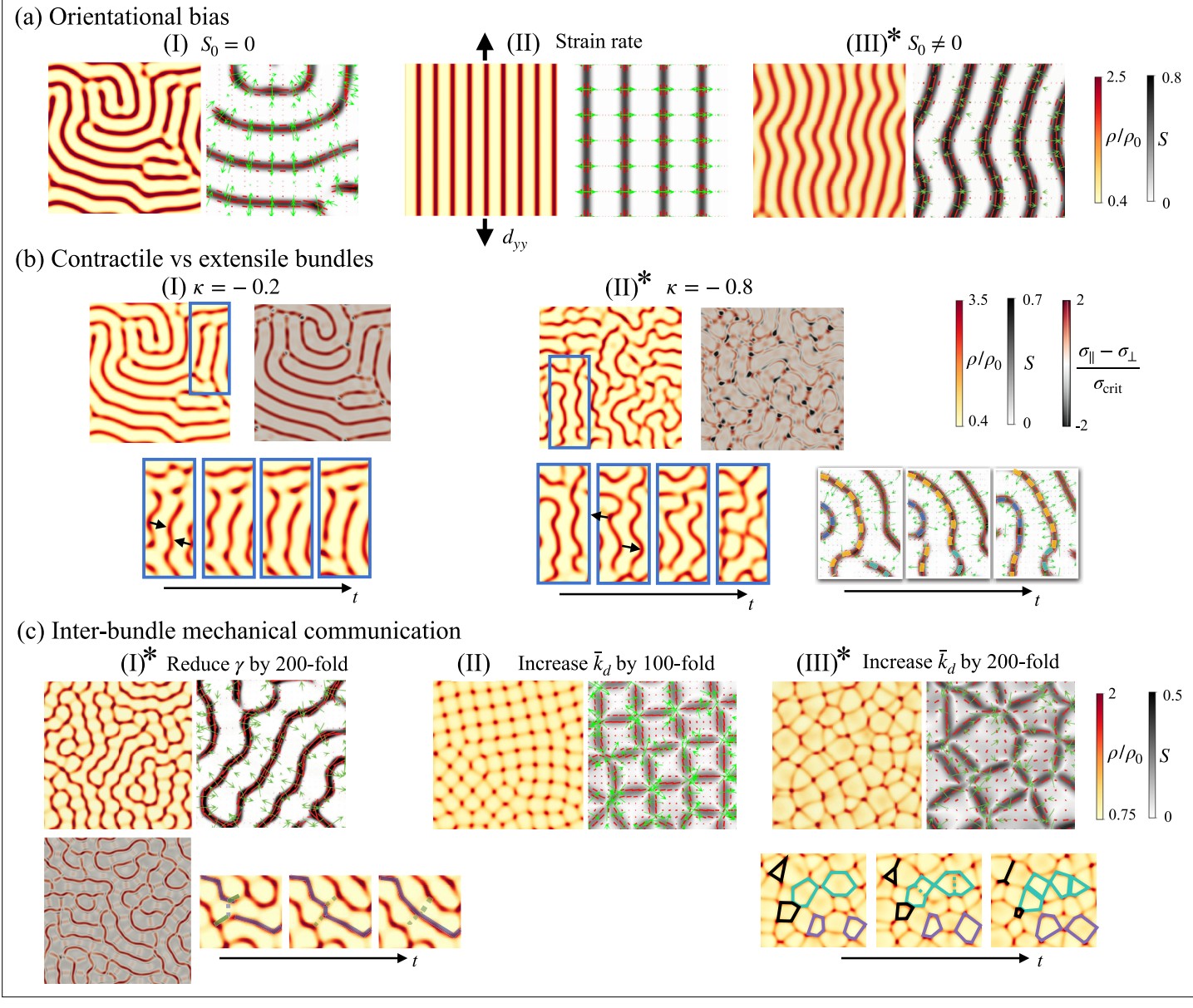

**Figure 4.** Control of nematic bundle pattern orientation, connectivity, and dynamics. (**a**) Effect of orientational bias. (**I**) A uniform isotropic gel self-organizes into a labyrinth pattern with defects. (**II**) A small background anisotropic strain rate efficiently orients nematic bundles. (**III**) A slight initial network alignment ($S_0 = 0.05$) orients bundles, which later lose stability, bend, and generate/anneal defects. See *Figure 4—video 1*. We recall that the nematic order parameter in the quiescent and uniform equilibrium state is $S_0 = 0$ if $c_0 \geq 0$ and $S_0 = \sqrt{-c_0/b}$ otherwise, with $c_0 = 2a - \rho_0 \lambda_\odot$. (**b**) Depending on active tension anisotropy, nematic bundles are contractile and straighten (I, $\kappa = -0.2$), leading to quasi-steady networks, or extensile and wrinkle (II, $\kappa = -0.8$), leading to bundle breaking and recombination, and persistently dynamic networks (III). See *Figure 4—video 2*. The contractility or extensibility of the nematic bundles can be appreciated in the maps of the difference between the stress along the nematic direction ($\sigma_\parallel$) and the stress perpendicular to it ($\sigma_\perp$), normalized by the constant $\sigma_{crit} = \lambda_{crit}\rho_0$. (**c**) Promoting mechanical interaction between bundles. (I) Dynamical pattern obtained by reducing friction, and thereby increasing $\bar{a}$, $\bar{b}$, $\bar{\lambda}_\odot$, and $\bar{k}_d$. Time sequence in the bottom indicates a typical reconfiguration event during which weak bundles (dashed) become strong ones (solid) and vice versa. (II) Nearly static pattern obtained increasing $\bar{k}_d$ (III) which becomes highly dynamic by further increasing $\bar{k}_d$. Time sequence in the bottom indicates the collapse (black), expansion (purple), and splitting (green) of cells in the network. See *Figure 4—video 3*. We indicate by ∗ dynamical patterns exhibiting spatiotemporal chaos.

The online version of this article includes the following video and figure supplement(s) for figure 4:

**Figure supplement 1.** Tension distribution along ($\sigma_\parallel$) and perpendicular to ($\sigma_\perp$) the dense nematic bundles.

**Figure 4—video 1.** Effect of orientational bias on fibrillar patterns.

https://elifesciences.org/articles/93097/figures#fig4video1

*Figure 4 continued on next page*

*Figure 4 continued*

**Figure 4—video 2.** Contractile vs extensile bundles as the magnitude of $\kappa$ increases, leading to active turbulence of fibrillar patterns.
https://elifesciences.org/articles/93097/figures#fig4video2

**Figure 4—video 3.** Changes in network geometry and dynamics following enhanced inter-bundle mechanical communication.
https://elifesciences.org/articles/93097/figures#fig4video3

contractile nematic active gel can self-organize into fibrillar patterns with mesoscale bundles that are either contractile or extensile depending on the parameter regime. This complex interplay between the contractility vs extensibility of the base material and that of the mesoscale nematic pattern resonates with recent observations where tissues made of contractile cells behave collectively as extensile nematic systems (*Balasubramaniam et al., 2022*; *Saw et al., 2017*).

Focusing on contractile bundles, we then examined a different parameter regime known to trigger sustained pattern dynamics. For isotropic gels, previous work has shown that reducing friction triggers chaotic dynamics as the distance between high-density regions, $\ell_{\text{pattern}} = 2\pi/\nu_{\text{crit}}$, becomes comparable to or smaller than the hydrodynamic lengthscale (*Hannezo et al., 2015*), thus enabling their mechanical interaction. In a model devoid of orientational order, reducing friction is equivalent to increasing $\bar{k}_d$. In our model, however, we can either reduce $\gamma$, which in nondimensional terms means increasing $\bar{k}_d$, $a$, $b$, and $\bar{\lambda}_\odot$ in concert, or increase $\bar{k}_d$ while leaving all other nondimensional parameters fixed. The first of these choices leads to dynamic and hierarchical networks with very dense and highly aligned bundles, which coexist with perpendicular weak bundles with much lower density enrichment and ordering. These two families of bundles enclose cells of isotropic and sparse gel (*Figure 4cI*). Junctions where two or more dense bundles meet are very unfavorable and short-lived, but junctions of two dense and a weak bundle are much more stable. Because bundles are mechanically coupled, the network actively reorganizes in events where dense bundles become weak bundles and vice versa (inset and *Figure 4—video 3I*). We note that here $\sigma_\parallel > \sigma_\perp$ (*Figure 4cI*, *Figure 4—figure supplement 1d*), and hence the persistent dynamics are unrelated to the previously described behavior of extensile bundles.

The second choice to favor mechanical interaction of bundles, increasing $\bar{k}_d$ only, leads to very different networks with high-density aster-like clusters interconnected by straight actin bundles. Because now $\bar{\lambda}_\odot$ is not particularly large, order is low at the core of these clusters, enabling high-valence networks where four bundles often meet at one cluster. For $\bar{k}_d = 10$, the network is stable and nearly crystalline (*Figure 4cII*), whereas for $\bar{k}_d = 20$, it becomes highly dynamical and pulsatile with frequent collapse of polygonal cells by fusion of neighboring actin clusters and their attached bundles (black polygons) and nucleation of new bundles within large low-density cells (dashed/solid green lines) (*Figure 4cIII*, *Figure 4—video 3III*). This architecture and dynamics resemble those of early *C. elegans* embryos (*Munro et al., 2004*), adherent epithelial cells treated with epidermal growth factor (*Jalal et al., 2019*), and mouse embryonic stem cells (*Gupta et al., 2015*). Recent active gel models accounting for RhoA signaling develop similar pulsatile behaviors in 2D but do not predict the orientational order of the spatiotemporal patterns of the actomyosin cortex (*Staddon et al., 2022*).

In summary, our theory maps how effective parameters of the actin gel control the active self-organization of a uniform and isotropic gel into a pattern of high-density nematic bundles embedded in a low-density isotropic matrix, including the activity threshold, the bundle spacing, orientation, connectivity, and dynamics.

## Microscopic origin of $\kappa < 0$ and $\lambda_\odot > 0$ through discrete network simulations

A central prediction of our model is that the self-organization of nematic bundles, the most prominent emerging organization in actin gels across cell types and lengthscales, requires that active tension perpendicular to nematic orientation is larger than along this direction ($\kappa < 0$, see *Figure 1c*), at least at the onset of pattern formation. This fact may seem counterintuitive in that dense nematic bundles are associated with large contractile tension along their axis, but as discussed in *Figure 4cI*, even if $\kappa < 0$, bundles can be contractile because of the interplay between active and viscous stresses. Another prediction of our model is that the active assembly of dense nematic bundles requires active alignment controlled by parameter $\lambda_\odot > 0$ (*Figure 3*). Being central to our conclusions, we then sought

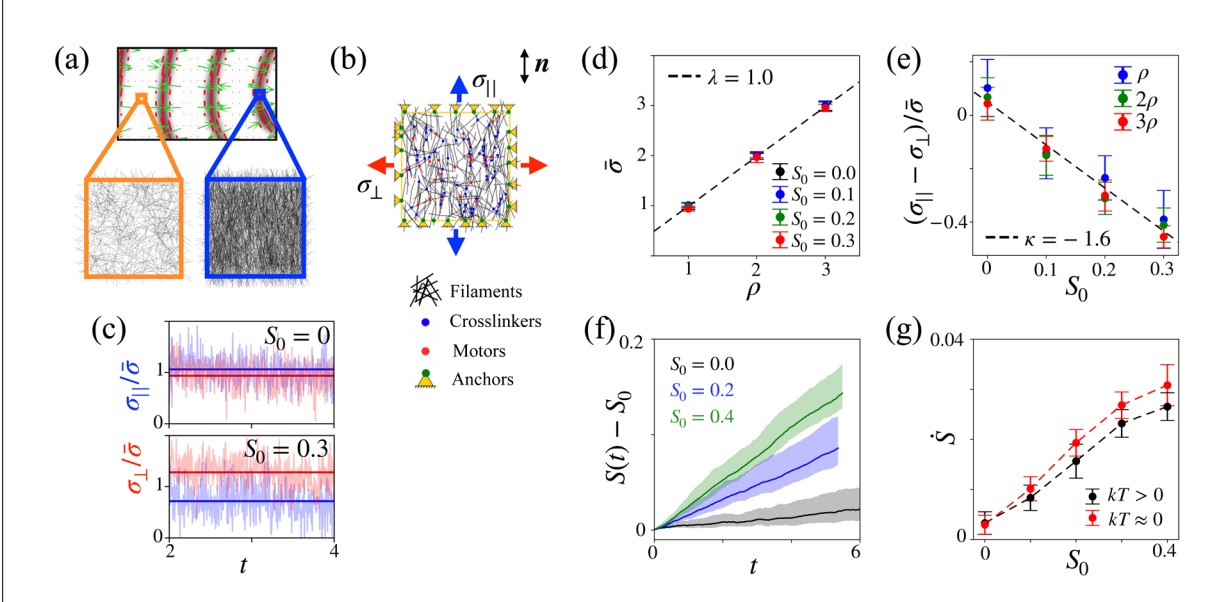

**Figure 5.** Assessment of activity parameters $\kappa$ and $\lambda_\odot$ through discrete network simulations. (**a**) Illustration of the computational domain of the discrete network as a uniform representative volume element of the gel. (**b**) Sketch of model ingredients and setup to compute tension along and perpendicular to the nematic direction. (**c**) Typical time signal for parallel and perpendicular tensions following addition of crosslinkers and motors (translucent lines) along with time average (solid lines) for isotropic and anisotropic networks. Tension is normalized by mean tension $\bar{\sigma} = (\sigma_\parallel + \sigma_\perp)/2$, computed from time averages and time by actin turnover time. (**d**) Mean tension as a function of network density for several nematic parameters $S_0$, where both quantities are normalized by their values for the lowest density. With this normalization, *Equation 11* predicts a linear dependence with slope $\lambda = 1$ (dashed line). Error bars span two standard deviations. (**e**) Deviatoric tension as a function of nematic order for different densities. The dashed line is a linear regression to simulation data. (**f**) Dynamics of nematic order in a periodic network following addition of crosslinkers and motors for three initial values of nematic order. (**g**) Rate of change of nematic order normalized by turnover rate as a function of initial nematic order at zero and finite temperature.

The online version of this article includes the following video and figure supplement(s) for figure 5:

**Figure supplement 1.** Detailed description of the discrete network simulations.

**Figure 5—video 1.** Illustration of simulation protocol to estimate tension anisotropy from discrete network simulations for an isotropic (left) and anisotropic (right) representative volume elements.
https://elifesciences.org/articles/93097/figures#fig5video1

**Figure 5—video 2.** Illustration of simulation protocol to estimate the generalized active tension conjugate to nematic order for an isotropic (left) and anisotropic (right) representative volume elements with periodic boundary conditions.
https://elifesciences.org/articles/93097/figures#fig5video2

to examine the plausibility of the conditions $\kappa < 0$ and $\lambda_\odot > 0$ of our phenomenological theory by performing discrete network simulations using the open-source cytoskeleton simulation suite cytosim (*Nedelec and Foethke, 2007*).

Similarly to reconstituted actomyosin gels, discrete network simulations of the actin cytoskeleton tend to irreversibly collapse into clumps (*Ennomani et al., 2016*), although proper tuning of model parameters can lead to sustained contractile states (*Mak et al., 2016*). However, to our knowledge, these models have not been able to reproduce the heterogeneous nonequilibrium contractile states involving sustained self-reinforcing flows underlying the pattern formation mechanism studied here. For this reason, rather than trying to reproduce the assembly and maintenance of patterns of dense nematic structures, we specifically focused on investigating whether the average behavior of the discrete network is compatible with the conditions $\kappa < 0$ and $\lambda_\odot > 0$ of our continuum theory. For this, we studied the buildup of tension and dynamics of nematic order in uniform representative volume elements of varying density and filament alignment after being brought out of equilibrium (*Figure 5a*). Each of our simulated volume elements can represent a uniform system prior to pattern formation, or a material point in the actomyosin gel.

We performed 2D simulations in which semiflexible filaments interact with crosslinkers and myosin motors, all of which undergo turnover and have a stoichiometry previously used to model the actin cytoskeleton (*Cortes et al., 2020*, *Figure 5b*). See Appendix D for a detailed description of the simulation protocol. Briefly, we modified cytosim to account for average orientational order in the simulation box, which we evaluated as a sample average of orientations over the ensemble of segments composing the filaments. We further introduced a nematic energy penalty in the network allowing us to restrain the average nematic order to a target value $S_0$.

We first prepared a system consisting only of randomly oriented actin fibers, imposed the desired orientational order $S_0$ using the nematic penalty, and equilibrated the system. In a first set of simulations, once $S_0$ was reached, we deactivated the nematic penalty and added crosslinkers and myosins, driving the system out of equilibrium. The free contraction of the system was prevented by the addition of anchors at the boundary of the representative volume element, which also allowed us to compute boundary forces and hence estimate the effective active tension along the nematic direction $\sigma_\parallel = \sigma_{ij} n_i n_j$ and perpendicular to it, $\sigma_\perp = \sigma_{ij} m_i m_j$ with $n_i m_i = 0$ and $m_i m_i = 1$ (*Figure 5b*).

Addition of crosslinkers and myosins leads to bundling of actin filaments at the microscale (*Figure 5—video 1*) distinct from the mesoscale fibrillar pattern formation emerging from the active gel model. It also leads to out-of-equilibrium tension as measured by the anchors. For isotropic networks ($S_0 = 0$), active tension is isotropic with $\sigma_\parallel \approx \sigma_\perp$. For anisotropic networks, however, we found that tension becomes anisotropic with $\sigma_\perp > \sigma_\parallel$ (*Figure 5c*).

We systematically characterized this behavior by varying initial orientational order and network density. According to our active gel model, *Equation 5*, *Equation 6*, *Equation 7*, in the absence of nematic gradients and flow, the tension components $\sigma_\parallel$ and $\sigma_\perp$ satisfy the following relations in terms of mean and deviatoric tensions

$$\bar{\sigma} = \left(\sigma_\parallel + \sigma_\perp\right)/2 = \lambda\rho \quad \text{and} \quad \left(\sigma_\parallel - \sigma_\perp\right)/\bar{\sigma} = \kappa S. \tag{11}$$

Remarkably, our discrete network simulations closely followed these relations (*Figure 5d, e*), which allowed us to estimate $\kappa \approx -1.6$. We robustly found that $\kappa < 0$ for perturbations of selected parameters of the discrete network model as long as turnover rates of crosslinkers and myosins were relatively fast.

We then wondered if the discrete network simulations could provide evidence for the orientational activity parameter in our theory, $\rho\lambda_\odot$. For a uniform system with nematic order along a given direction and ignoring the susceptibility parameter $b$, *Equation 8* becomes

$$\eta_{\text{rot}}\dot{S} + (2a - \rho\lambda_\odot)S + \mathcal{K}_S(S - S_0) = 0, \tag{12}$$

where $\eta_{\text{rot}}$ is the viscous drag of the filaments in the discrete network simulations, $a > 0$ the entropic tendency of the model to return to isotropy, $\rho\lambda_\odot$ the active forcing of nematic order resulting from crosslinkers and motors, and the last term accounts for the effect of the nematic penalty with coefficient $\mathcal{K}_S$. As a first test of this model, we started from an isotropic and periodic network and tracked the athermal dynamics of $S$ under the action of the nematic penalty in the absence of anchors, crosslinkers, and myosins. For $\lambda_\odot = 0$ and $a = 0$, *Equation 12* predicts an exponential relaxation given by $S(t) = S_0(1 - e^{-\mathcal{K}_S t/\eta_{\text{rot}}})$, which very closely matched the simulation data for different values of $S_0$ and for a single fitting parameter $\eta_{\text{rot}}$ (*Figure 5—figure supplement 1III*, *Figure 5—video 2*). We then deactivated the nematic penalty and added crosslinkers and motors, but not anchors, to track unconstrained dynamics of nematic order starting from different values of $S_0$. In agreement with the notion of an active force driving nematic order, we found that $S(t)$ monotonically increased (*Figure 5f*). More quantitatively, we tested the short-time prediction of *Equation 12*, $\eta_{\text{rot}}\dot{S} = (\rho\lambda_\odot - 2a)S_0$, by plotting $\dot{S}$ as estimated from our simulations, as a function of $S_0$ (*Figure 5g*). We found a nearly linear relation with positive slope, hence providing evidence for an active generalized force driving order. In agreement with the theory, in the athermal limit, the tendency to actively increase order is faster as the entropic tendency to disorder is absent ($a = 0$).

In summary, discrete network cytoskeletal simulations provide a microscopic justification for two key ingredients of our active gel theory, namely that nematic order elicits (1) anisotropic active tensions, which can be larger perpendicular to the nematic direction ($\kappa < 0$), and (2) active generalized forces driving further ordering.

## Discussion

We have developed a theory for the active self-organization of initially uniform and isotropic actin gels into various localized dense nematic architectures embedded in an isotropic matrix of low density. This model predicts a variety of emergent patterns involving asters, tactoids, and bands with perpendicular nematic organization. More importantly, it identifies a wide parameter space where the active gel spontaneously develops patterns of dense nematic bundles, the most prominent nematic architecture across scales and cell types. Contrary to previous works on instabilities and pattern formation in active nematics, the mechanism proposed here relies on the advective instability and compressible self-reinforcing flows typical of actomyosin gels. We have characterized how the activity threshold, spacing, geometry, connectivity, and dynamics of these patterns depend on effective active gel parameters. Because these mesoscale parameters depend on the composition and dynamics of the network at the molecular scale, our results portray actin gels as responsive and reconfigurable active materials with an intrinsic ability to assemble patterns of nematic bundles that cells can finely regulate.

We have further shown that the spontaneous tendency of the gel to assemble bundle patterns can be directed via subtle cues. Thus, a combination of biochemical control of actin dynamics along with geometric, mechanical, or biochemical guiding (*Gross et al., 2017*; *Burkart et al., 2022*) may explain the emergence and context-dependent organization of regular patterns of bundle networks, from subcellular to organism scales (*Senger et al., 2019*; *Tojkander et al., 2015*; *Wirshing and Cram, 2017*; *Öztürk-Çolak et al., 2016*; *Hotulainen and Lappalainen, 2006*; *Tee et al., 2015*; *Yolland et al., 2019*; *Jalal et al., 2019*; *Dinwiddie et al., 2014*; *Maroudas-Sacks et al., 2021*). Consistently, perturbations of myosin contractility have been shown to alter, disorder, or even prevent the formation patterns of parallel bundles in *C. elegans* (*Wirshing and Cram, 2017*), whereas perturbations of actin polymerization in *Drosophila* embryos impair the robust organization of actin bundle patterns at the cellular and organ scales by disrupting orientation and spacing, but not the intrinsic tendency of the actomyosin cytoskeleton to form patterns of parallel bundles (*Öztürk-Çolak et al., 2016*). In adherent cells, the patterns of dense nematic bundles presented here may act as precursors of stress fibers as the actomyosin cytoskeleton interacts with the focal adhesion machinery in a cellular domain with boundary conditions set by the polymerization velocity at the leading edge and the nucleus.

Our theory identifies two key requirements on activity parameters for the self-organization of patterns of nematic bundles, namely active tension anisotropy with larger tension perpendicular to the nematic direction and generalized active forces tending to increase nematic order. Complementarily to our phenomenological theory, we have examined the plausibility of these conditions with discrete network simulations of homogeneous representative volume elements of different density and orientational order, which have confirmed our constitutive assumptions. We expect, however, that in a different parameter regime, anisotropic tension may be larger along the nematic direction ($\kappa > 0$). For instance, once bundles are dense and maximally aligned, the ability of the active nematic gel to perform active tension perpendicular to the nematic direction may saturate, while myosin motors may contract the gel along the nematic direction more effectively. The regime studied here explains the initial assembly of dense nematic bundles, but not their maturation to become highly contractile or viscoelastic as demonstrated for stress fibers depending on different isoforms of nonmuscle myosin II or on actin regulators such as zyxin (*Weißenbruch et al., 2021*; *Oakes et al., 2017*). Our work thus suggests further experimental and computational work to establish a comprehensive mapping between molecular and mesoscale properties of the active gel, and how these properties control the emergent network architecture and mechanical properties.

## Materials and methods

The continuum simulations presented in *Figures 2–4* solve *Equations 1, 2, 8*, along with the constitutive *Equations 4–7*, on a periodic square domain using the finite element method. A detailed description of the implementation and the computer code is provided elsewhere (*Mirza et al., 2025*; *Mirza, 2025*).

The discrete network simulations in *Figure 5* were performed with an agent-based microscopic model of a crosslinked actomyosin network using the open-source cytoskeletal simulation suite cytosim (*Nedelec and Foethke, 2007*; *Nedelec, 2022*). We customized the source code to impose

and track nematic order in the system (*Pensalfini, 2025*). A detailed description of the model and the simulation parameters is provided in Appendix D.

## Acknowledgements

The authors acknowledge the support of the European Research Council (CoG-681434) and the Spanish Ministry for Science and Innovation (PID2019-110949GB-I00). WM acknowledges the La Caixa Fellowship and the European Union's Horizon 2020 research and innovation program under the Marie Skłodowska-Curie action (GA 713637). MP acknowledges the support from the Spanish Ministry of Science and Innovation & NextGenerationEU/PRTR (PCI2021-122049-2B). MA acknowledges the Generalitat de Catalunya (ICREA Academia prize for excellence in research). MDC acknowledges funding from the Spanish Ministry for Science and Innovation through the Juan de la Cierva Incorporación fellowship IJC2018-035270-I. IBEC and CIMNE are recipients of a Severo Ochoa Award of Excellence

## Additional information

### Funding

| Funder | Grant reference number | Author |
| --- | --- | --- |
| European Research Council | CoG-681434 | Marino Arroyo |
| Spanish National Plan for Scientific and Technical Research and Innovation | PID2019-110949GB-I00 | Marino Arroyo |
| Spanish National Plan for Scientific and Technical Research and Innovation | PCI2021-122049-2B | Marco Pensalfini |
| Spanish National Plan for Scientific and Technical Research and Innovation | IJC2018-035270-I | Marco De Corato |
| La Caixa Fellowship and the European Union's Horizon 2020 | GA 713637 | Waleed Mirza |

The funders had no role in study design, data collection and interpretation, or the decision to submit the work for publication.

### Author contributions

Waleed Mirza, Conceptualization, Software, Formal analysis, Investigation, Methodology, Writing – review and editing; Marco De Corato, Conceptualization, Formal analysis, Investigation, Methodology; Marco Pensalfini, Conceptualization, Software, Investigation, Methodology, Writing – review and editing; Guillermo Vilanova, Software, Supervision, Methodology; Alejandro Torres-Sánchez, Conceptualization, Software, Supervision, Investigation, Methodology, Writing – review and editing; Marino Arroyo, Conceptualization, Resources, Supervision, Funding acquisition, Investigation, Methodology, Writing - original draft, Project administration, Writing – review and editing

### Author ORCIDs

Waleed Mirza ⓘ https://orcid.org/0000-0002-5197-1371
Marco De Corato ⓘ https://orcid.org/0000-0002-9361-4794
Marco Pensalfini ⓘ https://orcid.org/0000-0003-3296-9388
Guillermo Vilanova ⓘ https://orcid.org/0000-0002-9650-0602
Alejandro Torres-Sánchez ⓘ https://orcid.org/0000-0002-4275-173X
Marino Arroyo ⓘ https://orcid.org/0000-0003-1647-940X

Reviewer #1 (Public review): https://doi.org/10.7554/eLife.93097.3.sa1
Reviewer #2 (Public review): https://doi.org/10.7554/eLife.93097.3.sa2

Reviewer #3 (Public review): https://doi.org/10.7554/eLife.93097.3.sa3
Author response https://doi.org/10.7554/eLife.93097.3.sa4

## Additional files

**Supplementary files**
MDAR checklist

**Data availability**
The current manuscript is a computational study, so no data have been generated for this manuscript. The code and input files are available at https://github.com/waleedmirzaPhD/actin_bundles (copy archived at *Mirza, 2025*) for the simulations based on the continuum theory and at https://gitlab.com/marco.pensalfini1/cytosim/-/tree/master/doc/papers/2024_Mirza_eLife (copy archived at *Pensalfini, 2025*) for discrete network simulations.

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

## Appendix A

### 1D reduced model

To perform the linear stability analysis, we reduce the theoretical model to 1D by considering the flow velocity $v(x, t)$ and the density field as $\rho(x, t)$ along the $x$-axis. The nematic order tensor is defined as $q_{ij} = S(n_i n_j - \delta_{ij}/2)$, where $\boldsymbol{n}$ is the local average filament orientation and $S$ is the local degree of alignment. Since it is traceless and symmetric, in general, it can be represented by two independent degrees of freedom, $q_{11} = -q_{22} = q_1$ and $q_{12} = q_{21} = q_2$. In the 1D model, we assume that $n$ is either along or perpendicular to $x$, and hence $q_{12} = q_{21} = 0$. We are thus left with one independent degree of freedom $q_{11} = -q_{22} = q$. Positive (negative) $q$ represents alignment along (perpendicular to) the $x$-axis. The Jaumann derivative of the nematic order parameter reduces in 1D to

$$\hat{q} = \frac{\partial q}{\partial t} + v \frac{\partial q}{\partial x}. \tag{A1}$$

Particularizing the general equations given in the main text, we present next the 1D governing equations pertinent to the linear stability analysis. Mass conservation reads

$$\frac{\partial \rho}{\partial t} + \frac{\partial}{\partial x}(\rho v) - D \frac{\partial^2 \rho}{\partial x^2} + k_d(\rho - \rho_0) = 0, \tag{A2}$$

balance of linear momentum of the active gel reduces to

$$\rho \gamma v = \frac{\partial \sigma}{\partial x}, \tag{A3}$$

where stress along $x$, $\sigma$ is given by

$$\sigma = \rho \left[ 4\eta \frac{\partial v}{\partial x} + \beta \hat{q} + \lambda (1 + \kappa q) - 2L \left( \frac{\partial q}{\partial x} \right)^2 \right]. \tag{A4}$$

The evolution of nematic order $q$ is governed in the 1D setting by

$$\eta_{\mathrm{rot}} \hat{q} + \frac{\beta}{2} \frac{\partial v}{\partial x} + \left( 2a - \rho \lambda_{\odot} + 4bq^2 \right) q - L \frac{\partial^2 q}{\partial x^2} - \frac{L}{\rho} \frac{\partial q}{\partial x} \frac{\partial \rho}{\partial x} = 0. \tag{A5}$$

For $c_0 = 2a - \rho_0 \lambda_{\odot} \geq 0$, the uniform steady state of the system is given by $\rho(x, t) = \rho_0$, $v(x, t) = 0$, and $q(x, t) = 0$. For $c_0 < 0$, the uniform steady state has spontaneous alignment given by $q(x, t) = q_0 = \pm \frac{1}{2} \sqrt{-c_0/b}$. Finally, we note that, for the dissipation to be positive, the material parameters need to satisfy an entropy production inequality $4\eta \eta_{\mathrm{rot}} - \beta^2 \geq 0$ in the reduced 1D model and $2\eta \eta_{\mathrm{rot}} - \beta^2 \geq 0$ in the 2D model (**Mirza et al., 2025**).

## Appendix B

### Nondimensionalization

We present next a nondimensionalization of the governing *Equations A2–A5* . We choose the hydrodynamical lengthscale $\ell_s = \sqrt{\eta/\gamma}$, above (below) which friction (viscosity) is the dominant dissipative mechanism in the active gel, and the timescale of diffusion over this lengthscale, $t_0 = \ell_s^2/D = \eta/(\gamma D)$. Using these characteristic scales, we nondimensionalize space and time as $\bar{x} = x/\ell_s$ and $\bar{t} = t/t_0$. We chose $\rho_0$ as a reference density and thus $\bar{\rho} = \rho/\rho_0$. Here and elsewhere, overbar denotes nondimensional quantities.

The dimensionless balance of mass reads

$$\frac{\partial \bar{\rho}}{\partial \bar{t}} + \bar{v}\frac{\partial \bar{\rho}}{\partial \bar{x}} + \bar{\rho}\frac{\partial \bar{v}}{\partial \bar{x}} - \frac{\partial^2 \bar{\rho}}{\partial \bar{x}^2} + \bar{k}_d(\bar{\rho} - 1) = 0, \tag{B1}$$

where $\bar{k}_d = k_d t_0$. The dimensionless balance of linear momentum reads

$$\bar{\rho}\,\bar{v} = \frac{\partial \bar{\sigma}}{\partial \bar{x}}, \tag{B2}$$

where $\bar{\sigma} = \sigma/\sigma_0$ and the characteristic stress is given by $\sigma_0 = \rho_0 \gamma D$. The nondimensional constitutive relation for the stress is

$$\bar{\sigma} = \bar{\rho}\left[4\frac{\partial \bar{v}}{\partial \bar{x}} + \bar{\beta}\bar{q} + \bar{\lambda}\left(1 + \kappa q\right) - 2\bar{L}\left(\frac{\partial q}{\partial \bar{x}}\right)^2\right], \tag{B3}$$

where $\bar{L} = L/(\gamma D\ell_s^2)$, $\bar{\beta} = \beta/\eta$, and $\bar{\lambda} = \lambda/(\gamma D)$. We note that $\kappa = \lambda_{\mathrm{aniso}}/\lambda$ and $q$ are already nondimensional. Finally, the dimensionless equation of generalized force conjugate to nematic order is given by

$$\bar{\eta}_{\mathrm{rot}}\dot{\bar{q}} + \frac{\bar{\beta}}{2}\frac{\partial \bar{v}}{\partial \bar{x}} + \left(2\bar{a} - \bar{\rho}\bar{\lambda}_\odot + 4\bar{b}q^2\right)q - \bar{L}\frac{\partial^2 q}{\partial \bar{x}^2} - \frac{\bar{L}}{\bar{\rho}}\frac{\partial q}{\partial \bar{x}}\frac{\partial \bar{\rho}}{\partial \bar{x}} = 0\,. \tag{B4}$$

where $\bar{\eta}_{\mathrm{rot}} = \eta_{\mathrm{rot}}/\eta$, $\bar{a} = a/\left(\gamma D\right)$, $\bar{b} = b/\left(\gamma D\right)$, and $\bar{\lambda}_\odot = \lambda_\odot \rho_0/(\gamma D)$. Inspection of *Equation B4* reveals the characteristic nematic lengthscale $\ell_q^2 = L/\left|2a - \lambda_\odot \rho_0\right|$ and the nematic relaxation timescale $t_q = (\ell_q^2 \eta_{\mathrm{rot}})/L = \eta_{\mathrm{rot}}/\left|2a - \lambda_\odot \rho\right|$.

## Appendix C

### Linear stability analysis

We perform next the linear stability analysis on the 1D model given by the dimensionless *Equations B1–B4*. For the sake of simplicity of notation, we omit overbars for the dimensionless quantities. We assess the stability of the homogeneous state given by $v(x,t) = 0$, $q(x,t) = q_0$, and $\rho(x,t) = \rho_0$ by examining the fate of infinitesimal perturbations $v(x,t) = \delta v(x,t)$, $q(x,t) = q_0 + \delta q(x,t)$, and $\rho(x,t) = \rho_0 + \delta\rho(x,t)$, where $|\delta v| \sim |\delta q| \sim |\delta\rho| \ll 1$. The control parameter in our stability analysis is the activity parameter $\lambda$.

Inserting the infinitesimally perturbed homogeneous states into the governing *Equations B1–B4* and ignoring terms that are quadratic in perturbations, the resulting balance of linear momentum equation is

$$\delta v = 4\frac{\partial^2 \delta v}{\partial x^2} + \beta\frac{\partial^2 \delta q}{\partial t \partial x} + \lambda(1 + \kappa q_0)\frac{\partial \delta\rho}{\partial x} + \lambda\kappa\frac{\partial \delta q}{\partial x}. \tag{C1}$$

The linearized form of *Equation B4* is

$$\eta_{\text{rot}}\frac{\partial \delta q}{\partial t} + \frac{\beta}{2}\frac{\partial \delta v}{\partial x} + \underbrace{\left[2a - \rho_0\lambda_\odot + 4bq_0^2 + 8bq_0^2\right]}_{G}\delta q - \lambda_\odot q_0 \,\delta\rho - L\frac{\partial^2 \delta q}{\partial x^2} = 0. \tag{C2}$$

If $c_0 = 2a - \rho_0\lambda_\odot > 0$, then $q_0 = 0$ and $G = c_0$. If $c_0 = 2a - \rho_0\lambda_\odot < 0$, then $q_0^2 = -c_0/(4b)$ and $G = 8bq_0^2$. The linearized balance of mass reads

$$\frac{\partial \delta\rho}{\partial t} + \frac{\partial \delta v}{\partial x} - \frac{\partial^2 \delta\rho}{\partial x^2} + k_d\delta\rho = 0. \tag{C3}$$

We decompose perturbations into a sum of Fourier terms, i.e., $\delta v = v'\, e^{\alpha t + \text{i}x\nu}$, $\delta\rho = \rho'\, e^{\alpha t + \text{i}x\nu}$, and $\delta q = q'\, e^{\alpha t + \text{i}x\nu}$, where $v'$, $q'$, and $\rho'$ are the amplitudes of the perturbations, $\alpha$ is the growth rate, and $\nu$ is the wavenumber.

Substituting the Fourier representation in *Equation C1*, we obtain

$$v'(1 + 4\nu^2) = q'\text{i}\nu(\beta\alpha + \lambda\kappa) + \rho'\text{i}\nu\lambda(1 + \kappa q_0). \tag{C4}$$

Likewise, *Equation C3* becomes

$$v' = \frac{\nu^2 + k_d + \alpha}{\nu}\text{i}\rho'. \tag{C5}$$

Substituting this equation into the Fourier representation of *Equation C2*, we obtain

$$q' = \frac{\beta(\alpha + \nu^2 + k_d) + 2\lambda_\odot q_0}{2\eta_{\text{rot}}\alpha + 2G + 2L\nu^2}\rho'. \tag{C6}$$

Finally, combining the three equations above, we obtain the dispersion equation, which, given the material parameters, relates wavenumber $\nu$ to growth rate $\alpha$ and takes the form

$$A(\nu)\,\alpha^2 + B(\nu)\,\alpha + C(\nu) = 0, \tag{C7}$$

where

$$\begin{aligned}
A(\nu) &= 2\eta_{\text{rot}}(4 + \nu^{-2}) - \beta^2, \\
B(\nu) &= (4 + \nu^{-2})(2G + 2L\nu^2) + 2\eta_{\text{rot}}\left[(4 + \nu^{-2})(\nu^2 + k_d) - \lambda(1 + \kappa q_0)\right] - \\
&\quad \beta\left[\beta(\nu^2 + k_d) + 2\lambda_\odot q_0 + \lambda\kappa\right], \\
C(\nu) &= \left[(4 + \nu^{-2})(\nu^2 + k_d) - \lambda(1 + \kappa q_0)\right](2G + 2L\nu^2) - \lambda\kappa\left[\beta(\nu^2 + k_d) + 2\lambda_\odot q_0\right],
\end{aligned}$$

The roots of *Equation C7* are given by $\alpha(\nu) = \left[-B(\nu) \pm \sqrt{B(\nu)^2 - 4A(\nu)C(\nu)}\right]/(2A(\nu))$. The system is linearly stable if perturbations decay, i.e., Real $(\alpha) < 0$, marginally stable if Real $(\alpha) = 0$, and unstable if Real $(\alpha) > 0$. We found that for all reasonable material parameters $\alpha(\nu)$ is real, and hence the critical condition is given by $\alpha = 0$, which recalling *Equation C7* implies that $C(\nu) = 0$. From this condition, we can express the activity parameter as

$$\lambda = \frac{(\nu^{-2} + 4)(\nu^2 + k_d)(2G + 2L\nu^2)}{(1 + \kappa q_0)(2G + 2L\nu^2) + \kappa\left[\beta(\nu^2 + k_d) + 2\lambda_\odot q_0\right]}. \tag{C8}$$

To find the critical wavenumber $\nu_{\text{crit}}$ and the associated critical activity parameter $\lambda_{\text{crit}} = \lambda(\nu_{\text{crit}})$, we minimize this expression with respect to $\nu$, or equivalently with respect to $x = \nu^2$, which requires that

$$\frac{\partial\lambda}{\partial x}(\nu_{\text{crit}}) = 0. \tag{C9}$$

## 1. Case A: $q_0 = 0$ and $L$ small

When $c_0 > 0$, the spontaneous order in the uniform steady-state vanishes and we have

$$\lambda = \frac{(x^{-1} + 4)(x + k_d)(2c_0 + 2Lx)}{2c_0 + 2Lx + \kappa\beta(x + k_d)}. \tag{C10}$$

Given a set of parameters, this equation can be numerically minimized to obtain the critical wavenumber and activity. To obtain workable explicit expressions, we first assume that $L = 0$, leading to

$$\lambda = \frac{(1 + 4x)(k_d + x)}{\delta x^2 + (1 + k_d\delta)x}, \tag{C11}$$

where

$$\delta = \frac{\kappa\beta}{2c_0}. \tag{C12}$$

From the condition $0 = \partial\lambda/\partial x$, we obtain

$$\nu_{\text{crit}}^2 = \frac{k_d\delta \pm \sqrt{k_d\left[4 + \delta(4k_d - 1)\right]}}{4 - \delta}. \tag{C13}$$

In the limit of $\delta \to 0$, we recover for the largest root the results $\nu_{\text{crit}}^2 = \nu_{\text{crit},0}^2 = \sqrt{k_d}/2$ and $\lambda_{\text{crit}} = \lambda_{\text{crit},0} = (1 + 2\sqrt{k_d})^2$ of an active gel model without nematic order (*Hannezo et al., 2015*). Taylor-expanding the equation above in terms of $\delta$, we obtain

$$\nu_{\text{crit}}^2 = \nu_{\text{crit},0}^2\left[1 + \frac{1}{8}\left(1 + 2\sqrt{k_d}\right)^2 \delta\right] + \mathcal{O}(\delta^2), \tag{C14}$$

and

$$\lambda_{\text{crit}} = \lambda_{\text{crit},0}\left[1 - \frac{\sqrt{k_d}}{2}\left(1 + 2\sqrt{k_d}\right)\delta\right] + \mathcal{O}(\delta^2). \tag{C15}$$

We note that these expressions are nondimensional. The dimensional expressions are reported in the main text.

## 2. Case B: $q_0 = 0$ and $c_0$ small

Using *Equation C10*, we examine the situation in which $\ell_s = 0.2$ approaches 0 but is positive so that $q_0 = 0$. *Equation C10* then becomes

$$\lambda = \frac{(1+4x)(k_d+x)}{(1+\delta)x+k_d\delta}, \tag{C16}$$

where now

$$\delta = \frac{\kappa\beta}{2L}. \tag{C17}$$

Following the same rationale as before, we find

$$\nu_{\text{crit}}^2 = \frac{-2k_d\delta + \sqrt{k_d\left[1+\delta(1-4k_d)\right]}}{2(1+\delta)}. \tag{C18}$$

Expanding up to linear order in $\delta$, we obtain

$$\nu_{\text{crit}}^2 = \nu_{\text{crit},0}^2\left[1 - \frac{1}{2}\left(1+2\sqrt{k_d}\right)^2\delta\right] + \mathcal{O}(\delta^2), \tag{C19}$$

and

$$\lambda_{\text{crit}} = \lambda_{\text{crit},0}\left[1 - \left(1+2\sqrt{k_d}\right)\delta\right] + \mathcal{O}(\delta^2). \tag{C20}$$

### 3. Case C: $q_0 \neq 0$ and $L$ small

Going back to *Equation C8* and taking $L = 0$ to obtain workable expressions, we obtain

$$\lambda = \frac{(1+4x)(x+k_d)}{\delta x^2 + (1+\tau+\delta k_d)x}, \tag{C21}$$

where

$$\delta = \frac{\kappa\beta}{16bq_0^2} \quad \text{and} \quad \tau = \kappa\left(\frac{\lambda_\odot}{8bq_0} + q_0\right). \tag{C22}$$

Minimization of $\lambda$ with respect to $x$ leads to

$$\nu_{\text{crit}}^2 = \frac{\delta k_d + \sqrt{\delta^2 k_d^2 + k_d(4+4\tau-\delta)(1+\tau+\delta k_d)}}{4+4\tau-\delta}, \tag{C23}$$

valid when $1 + \tau > 0$. Taylor-expanding the equation above in terms of $\delta$, we obtain

$$\nu_{\text{crit}}^2 = \nu_{\text{crit},0}^2\left[1 + \frac{1}{8}\frac{\left(1+2\sqrt{k_d}\right)^2}{1+\tau}\delta\right] + \mathcal{O}(\delta^2), \tag{C24}$$

and

$$\lambda_{\text{crit}} = \frac{\lambda_{\text{crit},0}}{1+\tau}\left[1 - \frac{\sqrt{k_d}}{2}\frac{1+2\sqrt{k_d}}{1+\tau}\delta\right] + \mathcal{O}(\delta^2). \tag{C25}$$

## Appendix D

## Choice of active gel model parameters in the main text

We consider reference model parameters for the actin cytoskeleton and variations about these values since they can be expected to change significantly between cells and conditions. We take a cortical viscosity of $\eta = 10^4$ Pa s, and hydrodynamic lengths in the order of $\ell_s = 0.2$ icrom (*Hannezo et al., 2015*; *Salbreux et al., 2009*), although we vary this parameter, as indicated in *Appendix 4—Tables 1 and 2*. Taking $k_d \sim 0.1\,\mathrm{s}^{-1}$ (*Yolland et al., 2019*; *Hannezo et al., 2015*; *Callan-Jones et al., 2008*) and $D \sim 0.04\,\mu\mathrm{m}^2\,\mathrm{s}^{-1}$ (*Yolland et al., 2019*; *Hannezo et al., 2015*), we obtain Damkölher lengths $\ell_D$ in the order of a micron. We consider $\eta_{\mathrm{rot}}/\eta \sim 1$ and $\beta/\eta \sim -0.2$, far from the threshold given by the entropy production inequality. We view $\lambda$ as the master activity parameter and take values a bit higher than the threshold for pattern formation, which depends on other material parameters as shown in our expressions for $\lambda_{\mathrm{crit}}$. We vary the nondimensional active tension anisotropy parameter $\kappa$. We choose the susceptibility parameters as $a/\eta_{\mathrm{rot}} = 5\,\mathrm{s}^{-1}$ and $b/\eta_{\mathrm{rot}} = 20\,\mathrm{s}^{-1}$ so that relaxation of nematic order is significantly faster than the turnover timescale and choose the Frank constant to be small so that $\ell_q$ is small compared to other lengthscales but not too small so that our computational grid can resolve it. The full list of material parameters for each figure and movie is given in *Appendix 4—table 1* and *Appendix 4—table 2*.

**Appendix 4—table 1.** Model parameters used in figures.

| Parameters | Figure 2 | Figure 4a | Figure 4b | Figure 4cI | Figure 4cII,III | Figure 3a | Figure 3b | Figure 3c |
|---|---|---|---|---|---|---|---|---|
| $\bar{k}_d$ | 0.1 | 0.1 | 0.1 | 20 | 10, 20 | 0.1 | 0.1 | 0.01 |
| $L$ | 1 | 1 | 1 | 1 | 1 | 1 | 1 | 1 |
| $\bar{c}_0$ | 4 | 4, 4, –0.05 | 4 | 200 | 4 | 4 | 4 | 0.4 |
| $b$ | 20 | 20 | 20 | 4000 | 20 | 20 | 20 | 2 |
| $\bar{\eta}_{rot}$ | 1 | 1 | 1 | 1 | 1 | 1 | 1 | 1 |
| $\bar{\beta}$ | –0.2 | –0.2 | –0.2 | –0.2 | –0.2 | –0.2 | $-\sqrt{2}$ | $-\sqrt{2}$ |
| $\bar{\lambda}_\odot$ | 6 | 6, 6, 10.05 | 6 | 1800 | 6 | 0 | 0 | 0 |
| $\bar{\lambda}$ | $1.3\bar{\lambda}_{crit}$ | $1.3\bar{\lambda}_{crit}$ | $1.3\bar{\lambda}_{crit}$ | $1.3\bar{\lambda}_{crit}$ | $1.3\bar{\lambda}_{crit}$ | $1.3\bar{\lambda}_{crit}$ | $1.3\bar{\lambda}_{crit}$ | $1.3\bar{\lambda}_{crit}$ |
| $\kappa$ | [–0.8, 0.8] | –0.2 | –0.2, –0.8 | –0.2 | –0.2 | –1.5, –0.75, 0, 0.75 | –1.5, –0.75, 0, 0.75 | –0.75, –0.375, 0, 0.75 |
| Domain Size $\times\, \nu_{crit}/(2\pi)$ | 8 | 8 | 8 | 8 | 8 | 8 | 8 | 8 |

**Appendix 4—table 2.** Model parameters used in movies that do not directly reproduce figures.

| Parameters | *Figure 2—video 1* | *Figure 3—video 1 (top)* | *Figure 3—video 1 (bottom)* | *Figure 4—video 2* |
|---|---|---|---|---|
| $\bar{k}_d$ | 0.01 | 0.1 | 0.1 | 0.1 |
| $L$ | 1 | 1 | 1 | 1 |
| $\bar{c}_0$ | 0.4 | 4 | 4 | 4 |
| $b$ | 2 | 20 | 20 | 20 |
| $\bar{\eta}_{rot}$ | 0 | 1 | 1 | 1 |
| $\bar{\beta}$ | 0 | –0.2, 0 | $-\sqrt{2}$, –0.2 | –0.2 |
| $\bar{\lambda}_\odot$ | 0 | 6 | 6 | 6 |
| $\bar{\lambda}$ | $1.3\,\bar{\lambda}_{crit}$ | $1.3\,\bar{\lambda}_{crit}$ | $1.3\,\bar{\lambda}_{crit}$ | $1.3\,\bar{\lambda}_{crit}$ |
| $\kappa$ | 0 | –0.2 | 0.2 | –0.2, –0.5, –0.8 |
| Domain Size $\times\, \nu_{crit}/(2\pi)$ | 8 | 8 | 8 | 8 |

# Appendix E

## Discrete network simulations

To ascertain the microstructural origin of the anisotropic active tensions underlying the formation of stress fibers and provide evidence for the orientational activity parameter ($\rho\lambda_\odot$) included in our theory, we establish an agent-based microscopic model of a crosslinked actomyosin network using the open-source cytoskeletal simulation suite cytosim (*Nedelec and Foethke, 2007*; *Nedelec, 2022*). We customized the source code to impose and track nematic order in the system (*Pensalfini, 2025*). Below, we provide a concise description of the modeling approach and an overview of the simulation parameters.

### 1. Model description

Actin filaments are represented through a Langevin equation that recreates the Brownian motion of a thermal system with temperature $T$ (thermal energy: $kT$, where $k = 1.38 \times 10^{-5}$ pN µm K$^{-1}$ is Boltzmann's constant) and includes bending elasticity, as well as external forces (e.g. those applied by crosslinkers and myosin motors). The filaments are immersed in a medium of viscosity $\nu$ and the motion of their points is described by

$$d\boldsymbol{X} = \mu\boldsymbol{F}(\boldsymbol{X}, t)dt + d\boldsymbol{B}(t),$$
(E1)

where $\boldsymbol{F}(\boldsymbol{X}, t)$ contains the forces acting on the points at time $t$, $d\boldsymbol{B}(t)$ is a stochastic nondifferentiable function of time summarizing the random molecular collisions that lead to Brownian motions, and the matrix $\mu$ contains the mobility coefficients of the particles that constitute the system.

The simulated system comprises $N_A$ actin filaments, which are modeled as inextensible objects with bending rigidity $\kappa_A$ and individual filament length $\ell_A$; both quantities are uniform across the system. Following a previously proposed approach (*Belmonte et al., 2017*), we represent actin turnover by randomly selecting and completely removing $N_t$ filaments every $N_s$ time steps and replacing them with an equal number of new ones that are placed at randomly chosen locations and have the same orientations as the removed filaments. The values of $N_t$ and $N_s$ are chosen to ensure that a given fraction of the total filaments in the system be replaced per unit of time, according to a turnover rate $r_A = N_t/(N_s N_A)$.

To control the nematic characteristics of the filament network, we introduce a system-wide restraining energy of stiffness $\mathcal{K}_S$ that penalizes deviations from a target orientational order parameter $S_0$, measured with respect to a director, $\hat{\boldsymbol{n}}$, that is aligned with the Cartesian basis vector $\hat{\boldsymbol{e}}_2$ (see *Figure 5—figure supplement 1I(a)*),

$$\mathrm{E}_S(\boldsymbol{X}) = \frac{\mathcal{K}_S}{2}\left|\boldsymbol{q}^s(\boldsymbol{X}) - \frac{S_0}{2}\begin{pmatrix} -1 & 0 \\ 0 & 1 \end{pmatrix}\right|^2.$$
(E2)

In this equation, we evaluate the average nematic order of the simulation box as a sample average given as $q^s_{ij} = \sum_{I=1}^{N}\left(m^I_i m^I_j - \delta_{ij}/2\right)/N$ over the ensemble of $N$ segments that constitute the filament network and $\boldsymbol{m}^I$ denotes the unit vector corresponding to the $I$th segment in the system. The energy defined by *Equation E2* results in restraining forces that act on the particles comprising the actin filaments, such that a particle of position $\boldsymbol{X}^P$ is subjected to a force

$$\boldsymbol{f}^P = \boldsymbol{f}\left(\boldsymbol{X}^P\right) = -\frac{\partial E_S(\boldsymbol{X})}{\partial \boldsymbol{X}^P} = -\frac{\partial E_S(\boldsymbol{X})}{\partial \boldsymbol{q}^s(\boldsymbol{X})}\frac{\partial \boldsymbol{q}^s(\boldsymbol{X})}{\partial \boldsymbol{X}^P} = -\mathcal{K}_S\left[\boldsymbol{q}^s(\boldsymbol{X}) - \frac{S_0}{2}\begin{pmatrix} -1 & 0 \\ 0 & 1 \end{pmatrix}\right]\frac{\partial \boldsymbol{q}^s(\boldsymbol{X})}{\partial \boldsymbol{X}^P}.$$
(E3)

In order to write the derivative $\partial\boldsymbol{q}^s(\boldsymbol{X})/\partial\boldsymbol{X}^P$ explicitly, we denote with $\boldsymbol{m}^{P,-} = \left(\boldsymbol{X}^P - \boldsymbol{X}^{P-1}\right)/\left|\boldsymbol{X}^P - \boldsymbol{X}^{P-1}\right|$ and $\boldsymbol{m}^{P,+} = \left(\boldsymbol{X}^{P+1} - \boldsymbol{X}^P\right)/\left|\boldsymbol{X}^{P+1} - \boldsymbol{X}^P\right|$ the unit vectors that correspond to the filament segments extending from point $\boldsymbol{X}^P$ toward the head (+) and the tail (−) of the filament (see *Figure 5—figure supplement 1I(b)*). In the absence of filament bifurcations, which are not present in our model, $\boldsymbol{m}^{P,+}$ and $\boldsymbol{m}^{P,-}$ are the only unit vectors in the network that depend on $\boldsymbol{X}^P$. Thus, it is immediate to simplify the derivative on the right-hand side of *Equation E3* and express the restraining force acting on the $P$th particle as

$$f^P = -\frac{\mathcal{K}_S}{N}\left[q_{ij}^s - \frac{S_0}{2}\begin{pmatrix} -1 & 0 \\ 0 & 1 \end{pmatrix}\right]\begin{pmatrix} \frac{\partial m_i^{P,-}}{\partial X_i^P}m_j^{P,-} + m_i^{P,-}\frac{\partial m_j^{P,-}}{\partial X_i^P} + \frac{\partial m_i^{P,+}}{\partial X_i^P}m_j^{P,+} + m_i^{P,+}\frac{\partial m_j^{P,+}}{\partial X_i^P} \\ \frac{\partial m_i^{P,-}}{\partial X_j^P}m_j^{P,-} + m_i^{P,-}\frac{\partial m_j^{P,-}}{\partial X_j^P} + \frac{\partial m_i^{P,+}}{\partial X_j^P}m_j^{P,+} + m_i^{P,+}\frac{\partial m_j^{P,+}}{\partial X_j^P} \end{pmatrix}.$$

(E4)

We shall also note that *Equation E4* is simplified when particle P coincides with the head or the tail of a filament, since either $m^{P,+}$ or $m^{P,-}$ will not exist. For a filament head, the restraining energy introduced in *Equation E2* will thus result in the force

$$f_{\text{head}}^P = -\frac{\mathcal{K}_S}{N}\left[q_{ij}^s - \frac{S_0}{2}\begin{pmatrix} -1 & 0 \\ 0 & 1 \end{pmatrix}\right]\begin{pmatrix} \frac{\partial m_i^{P,-}}{\partial X_i^P}m_j^{P,-} + m_i^{P,-}\frac{\partial m_j^{P,-}}{\partial X_i^P} \\ \frac{\partial m_i^{P,-}}{\partial X_j^P}m_j^{P,-} + m_i^{P,-}\frac{\partial m_j^{P,-}}{\partial X_j^P} \end{pmatrix}.$$

(E5)

Similarly, the force acting at a filament tail will be

$$f_{\text{tail}}^P = -\frac{\mathcal{K}_S}{N}\left[q_{ij}^s - \frac{S_0}{2}\begin{pmatrix} -1 & 0 \\ 0 & 1 \end{pmatrix}\right]\begin{pmatrix} \frac{\partial m_i^{P,+}}{\partial X_i^P}m_j^{P,+} + m_i^{P,+}\frac{\partial m_j^{P,+}}{\partial X_i^P} \\ \frac{\partial m_i^{P,+}}{\partial X_j^P}m_j^{P,+} + m_i^{P,+}\frac{\partial m_j^{P,+}}{\partial X_j^P} \end{pmatrix}.$$

(E6)

Finally, assuming that each actin filament is represented by segments of constant length $s_A$, the derivatives of the unit vectors appearing in *Equations E4–E6* are obtained directly from the definitions of $m^{P,-}$ and $m^{P,+}$ as

$$\begin{aligned}
\frac{\partial m_i^{P,-}}{\partial X_i^P} &= \frac{\left(X_j^P - X_j^{P-1}\right)^2}{s_A^3}, & \frac{\partial m_j^{P,-}}{\partial X_i^P} &= -\frac{\left(X_i^P - X_i^{P-1}\right)\left(X_j^P - X_j^{P-1}\right)}{s_A^3}, \\
\frac{\partial m_i^{P,-}}{\partial X_j^P} &= -\frac{\left(X_i^P - X_i^{P-1}\right)\left(X_j^P - X_j^{P-1}\right)}{s_A^3}, & \frac{\partial m_j^{P,-}}{\partial x_j^P} &= \frac{\left(X_i^P - X_i^{P-1}\right)^2}{s_A^3}, \\
\frac{\partial m_i^{P,+}}{\partial X_i^P} &= -\frac{\left(X_j^{P+1} - X_j^P\right)^2}{s_A^3}, & \frac{\partial m_j^{P,+}}{\partial X_i^P} &= \frac{\left(X_i^{P+1} - X_i^P\right)\left(X_j^{P+1} - X_j^P\right)}{s_A^3}, \\
\frac{\partial m_i^{P,+}}{\partial X_j^P} &= \frac{\left(X_i^{P+1} - X_i^P\right)\left(X_j^{P+1} - X_j^P\right)}{s_A^3}, & \frac{\partial m_j^{P,+}}{\partial X_j^P} &= -\frac{\left(X_i^{P+1} - X_i^P\right)^2}{s_A^3}.
\end{aligned}$$

(E7)

Crosslinkers and myosin motors are modeled as Hookean springs with finite stiffnesses $K_X$ and $K_M$ and resting lengths $\ell_X$ and $\ell_M$, respectively. For simplicity, the concentration of unattached species is assumed to be uniform across the modeling space and their diffusion is thus not simulated explicitly. The ends of the springs can bind to any discrete location along the filament segments, as long as they fall within finite binding ranges denoted as $d_X^b$ and $d_M^b$. When multiple filament locations fall within the binding range, the springs attach to the closest point on the filament and apply a force that depends linearly on their elongation ($u_X$ and $u_M$)

$$\begin{aligned}
f_X &= K_X\left(u_X - \ell_X\right), \\
f_M &= K_M\left(u_M - \ell_M\right).
\end{aligned}$$

(E8)

Binding of the species is modeled as a purely stochastic event whose time of occurrence follows an exponential distribution. The expected values for crosslinker and motor binding are $1/r_X^b$ and $1/r_M^b$, respectively, and $r_X^b$ and $r_M^b$ being the corresponding binding rates. Unbinding from a filament is not always purely stochastic but follows a rate that can increase when the springs are loaded, according to a relationship known as Bell's law (*Bell, 1978*). Denoting the base unbinding rates for crosslinkers and motors as $r_X^{u,0}$ and $r_M^{u,0}$, the effective rates under a force of magnitude $f$ are expressed as

$$r_X^u = r_X^{u,0} e^{f/f_X^u},$$
$$r_M^u = r_M^{u,0} e^{f/f_M^u},$$

(E9)

where $f_X^u$ and $f_M^u$ are constant parameters associated with the bound state. A specific feature of motors is that they can 'walk' on actin filaments by displacing their attachment points without necessarily having to unbind; this effect can be modified by load application. Denoting with $v_M^{max}$ the speed of the motor in the absence of external loads, the effective speed is given by

$$v_M = v_M^{max} (1 + f/f_M^{stall}),$$

(E10)

where $f_M^{stall}$ is the stall force, which controls the slope of the speed-force relationship. By convention, we consider that positive (negative) speed values correspond to a motion toward the head (tail) of a filament.

## 2. Quantification of network-scale active tensions

To determine anisotropic active tensions arising when the network is driven out-of-equilibrium, we examine a square representative volume element of crosslinked actomyosin network at room temperature ($kT = 0.0042$ pN μm). The modeled systems feature an edge length of 5 μm, a filament surface density $\rho_A$, a crosslinker density $\rho_X = 15.4$ (actin μm)$^{-1}$, and a myosin motor density $\rho_M = 0.8$ (actin μm)$^{-1}$. The actin density $\rho_A$ is varied throughout the study and takes the following values: $\rho_A = (78; 156; 234)$ (actin μm)/μm$^2$, which we denote as $\rho$, $2\rho$, and $3\rho$. All simulation parameters adopted in this study are indicated in *Appendix 5—Tables 1–4* where we also provide an overview of the values used in previously proposed microstructural models that are comparable to the ones developed here. Throughout the simulations, we adopt a time step of 5 ms and a data acquisition frequency of 40 s$^{-1}$, i.e., 1 every 5 time steps is written to output for further data processing.

The filaments are initially seeded uniformly throughout the modeling space and without any orientational bias (*Figure 5—figure supplement 1II(a)*) followed by equilibration in the presence of the restraining energy defined by *Equation E2*, which acts with a stiffness $\mathcal{K}_S = 5000$ pN/μm to induce a nematic orientation characterized by the ordering parameter $S_0$ (see *Figure 5—figure supplement 1II(b)*). No crosslinkers or myosin motors are present at this stage. After 5 s, *Figure 5—figure supplement 1II(c)*, the filaments are clamped to the system's boundary using $N_H$ anchors. These objects, whose spatial position is fixed, are akin to crosslinkers but have spring stiffness $K_H = 5000$ pN/μm, zero resting length, a binding range $d_H^b = 5$ nm, and a binding rate $r_H^b = 100$ s$^{-1}$. Anchor unbinding is completely hindered by setting $r_H^{u,0} = 0$ s$^{-1}$ and $f_H^u = \infty$. The quantity $N_H$ is defined by the number of filaments that cross the system's boundaries. Indicating with $N_H^L$, $N_H^R$, $N_H^B$, and $N_H^T$ the number of anchoring objects located on the left, right, bottom, and top edge of the system, it follows that $N_H = N_H^L + N_H^R + N_H^B + N_H^T$.

After having equilibrated the network for an additional 2.5 s, we deactivate the restraining potential by setting $\mathcal{K}_S = 0$ and introduce crosslinkers and motors in the system (*Figure 5—figure supplement 1II(d)*), which we let interact with the filaments for 12.5 s in order to drive the system out-of-equilibrium. This results in reaction forces being applied at the anchoring points. At each time step, the total average tensions acting on the system edges that are oriented parallel and perpendicular to the nematic director $\hat{n}$, $\sigma_\parallel$, and $\sigma_\perp$ are measured as

$$\sigma_\parallel = \frac{1}{2L} \left( \sum_{I=1}^{N_H^T} \boldsymbol{F}_I^T \cdot \hat{\boldsymbol{e}}_2 - \sum_{I=1}^{N_H^B} \boldsymbol{F}_I^B \cdot \hat{\boldsymbol{e}}_2 \right),$$
$$\sigma_\perp = \frac{1}{2L} \left( \sum_{I=1}^{N_H^R} \boldsymbol{F}_I^R \cdot \hat{\boldsymbol{e}}_1 - \sum_{I=1}^{N_H^L} \boldsymbol{F}_I^L \cdot \hat{\boldsymbol{e}}_1 \right).$$

(E11)

Measuring tensions from $t = 10$ to $t = 20$ (*Figure 5—figure supplement 1II(e, f)*), allows us to quantify tension anisotropy using the deviatoric tension normalized by the mean tension

$$\xi = 2 \frac{< \sigma_\parallel - \sigma_\perp >}{< \sigma_\parallel + \sigma_\perp >},$$

(E12)

where $< \cdot >$ denotes the time average over the considered period of interest. Finally, quantifying the above ratio for 12 sets of $n = 16$ model realizations that correspond to values of $S_0 = (0.0; 0.1; 0.2; 0.3)$ and $\rho_A = (\rho; 2\rho; 3\rho)$ allows us to determine the mesoscale activity parameter $\kappa$ by fitting the $\xi$ vs $S_0$ relation (see **Figure 5e**). To this end, we leverage the function 'stats.linregress' that is included in the Python package 'SciPy' (**Jones et al., 2001**).

## 3. Quantification of the orientational activity parameter

To provide evidence for the orientational activity parameter in our theory ($\rho\lambda_\odot$), we examine the behavior of a periodic square representative volume element of crosslinked actomyosin driven out-of-equilibrium. The absence of anchors in these systems allows us to track the unconstrained nematic order dynamics. We limit our analysis to models with actin density $\rho_A = 2\rho$ and adopt the same simulation parameters as in the section Quantification of network-scale active tensions, unless explicitly stated otherwise.

To highlight the contribution of the restraining potential in **Equation E2**, we initially focus on athermal systems ($kT \approx 0$). After seeding $N_A$ actin filaments uniformly throughout the modeling space and without any orientational bias (**Figure 5—figure supplement 1III(a)**), we let the restraining potential reorient the system for 2.5 s in the absence of crosslinkers and myosin motors to reach an ordering parameter, $S_0$ (**Figure 5—figure supplement 1III(b)**). Following the first 2.5 s of athermal simulation, we increase the temperature to its standard value ($kT = 0.0042$ pN μm), deactivate the restraining potential by setting $\mathcal{K}_S = 0$, and add crosslinkers and myosin motors to the system, which we let interact with the filaments for 30 s in order to drive the system out-of-equilibrium (**Figure 5—figure supplement 1III(c,d)**). We observe that $S(t)$ increases monotonically, allowing us to estimate $\dot{S}$ according to a linear least-squares fit, as implemented in the Python 'SciPy' function 'stats.linregress' (**Jones et al., 2001**). We notice that in **Figure 5—figure supplement 1III(a, b)**, the time evolution of $S$ is approximately exponential (**Figure 5—figure supplement 1III(e)**), as predicted in the main text. This allows us to leverage our discrete network simulations to estimate the model parameter $\eta_{\text{rot}} \approx 1583$ pN μm s using the nonlinear least-squares approach provided by the function 'optimize.curve_fit' that is included in the Python package 'SciPy' (**Jones et al., 2001**). Note that the value of $\eta_{\text{rot}}$ is determined by fitting one single evolution curve for $S(t)/S_0$, which is obtained by averaging the system dynamics resulting from the 4 sets of $n = 16$ model realizations that correspond to values of $S_0 = (0.1; 0.2; 0.3; 0.4)$.

Finally, we plot our simulation-based approximation of $\dot{S}$ for several values of $S_0$, which results in an almost linear relationship with positive slope, implying that $\rho\lambda_\odot > 2a$ (see **Figure 5g**). Moreover, considering fully athermal simulations (i.e. simulations in which the temperature is not increased after reaching the ordering parameter $S_0$) results in an linear relation between $\dot{S}$ and $S_0$ with slightly larger slope. This finding agrees with the predictions of our theory for $a > 0$ and demonstrates that temperature, which entropically drives the network toward isotropy, opposes the effect of the orientational activity parameter $\rho\lambda_\odot$.

**Appendix 5—table 1.** Global parameters adopted in this study and in previous microstructural models that used cytosim.

| Reference | | Geometry and size (μm) | $kT$ (pN μm) | Time step (S) | $\nu$ (Pa s) |
|---|---|---|---|---|---|
| Present work | | Square: $\ell = 5$ | 0 or 0.0042 | 0.005 | 1 |
| | MM1 | Rectangle: $2 \times 0.2$ | 0.0042 | 0.001 | 1 |
| | MM4 | Rectangle: $9.424 \times 1$ | 0.0042 | 0.001 | 1 |
| *Cortes et al., 2020* | MM3 and MM5 | Circle: $R = 1.5$ | 0.0042 | 0.001 | 1 |
| Cortex simulations in *Wollrab et al., 2019* | | Square: $L = 8$ | 0.0042 | 0.002 | 0.18 |
| *Bun et al., 2018* | | Circle: $R = 10$ | 0.0042 | 0.01 or 0.1 | 0.3 |
| Model with turnover in *Belmonte et al., 2017* | | Square: $L = 16$ | 0.0042 | 0.001 | 0.1 |
| *Descovich et al., 2018* | | Circle: $R = 15$ | 0.0042 | 0.002 | 1 |

*Appendix 5—table 1 Continued on next page*

*Appendix 5—table 1 Continued*

| Reference | Geometry and size (µm) | $kT$ (pN µm) | Time step (S) | $\nu$ (Pa s) |
|---|---|---|---|---|
| *Ding et al., 2017* | Circle: $R = 10$ | 0.0042 | 0.001 | 0.1 |
| *Ennomani et al., 2016* | Ring: $R = 4.5$, $t \approx 0.5$ | 0.0042 | 0.01 | 0.18 |

**Appendix 5—table 2.** Actin filament parameters adopted in this study and in previous microstructural models that used cytosim.

| Reference | | $\ell_A$ (µm) | $\rho_A$ (µm/µm²) | $s_A$ (µm) | $\kappa_A$ (pN µm²) | $r_A$ (%/s) |
|---|---|---|---|---|---|---|
| Present work | | 1.3 | 78, 156, or 234 | 0.2 | 0.1 | 20 |
| | MM1 | 2 | 1800 | 0.05–0.1 | 0.06 | 0 |
| | MM3 | 1.3±0.3 | 5.09–81.5 | 0.05–0.1 | 0.06 | 0 |
| | MM4 | 1.3±0.3 | 38.2–61.1 | 0.05–0.1 | 0.06 | 0 |
| *Cortes et al., 2020* | MM5 | 1.3±0.3 | 50.9–81.5 | 0.05–0.1 | 0.06 | 0 |
| Cortex simulations in *Wollrab et al., 2019* | | 0.1–4.0 | 5.5–218.8 | 0.1–0.4 | 0.075 | 0 |
| *Bun et al., 2018* | | 1.5 | 23.9 | 0.15 | 0.07 | 0 |
| Model with turnover in *Belmonte et al., 2017* | | 5 | 27.3 | 0.1–0.2 | 0.075 | 1.1–18.3 |
| *Descovich et al., 2018* | | 1.3±0.3 | 3.4–5.4 | 0.1 | 0.06 | 0 |
| *Ding et al., 2017* | | 2.2 | 35 | – | 0.05 | 0 |
| *Ennomani et al., 2016* | | 0.95–1.75 | 117.2–154.2 | – | 0.042–0.063 | 0 |

**Appendix 5—table 3.** Myosin motor parameters adopted in this study and in previous microstructural models that used cytosim.

| Reference | $\rho_M$ (1/µm actin) | $K_M$ (pN/µm) | $\ell_M$ (µm) | $r_M^b$ (1/s) | $d_M^b$ (µm) | $r_M^{u,0}$ (1/s) | $f_M^u$ (pN) | $f_M^{stall}$ (pN) | $v_M^{max}$ (µm/s) |
|---|---|---|---|---|---|---|---|---|---|
| Present work | 0.8 | 250 | 0 | 50 | 0.02 | 50 | ∞ | 6 | 0.3 |
| *Weißenbruch et al., 2021* | 0.5 | 100 | 0.3 | 0.2–3.6 | 0.06–0.12 | 0.8–1.71 | 5 | 3.85–15 | 0.137–0.6 |
| *Weißenbruch et al., 2021* | 0.6–1.0 | 100 | 0.3 | 0.2–3.6 | 0.06–0.12 | 0.8–1.71 | 5 | 3.85–15 | 0.137–0.6 |
| Cortex simulations in *Wollrab et al., 2019* | 2–64 | 100 | 0 | 10 | 0.01 | 0.5 | ∞ | 4 | 2 |
| *Bun et al., 2018* | 2.7 | 250 | 0.01 | 10 | 0.01 | 0.1 | 3 | 6 | 0.02 |
| Model with turnover in *Belmonte et al., 2017* | 3.2 | 500 | 0 | 10 | 0.01 | 0.3 | ∞ | 6 | 0.2 |
| *Descovich et al., 2018* | 0.5–0.8 | 1400 | 0.32 | 0.2 | 0.33 | 0.3 | 3.85 | 24.5 | 0.1 |
| *Ding et al., 2017* | 0 or 5.8 | 250 | 0 | 10 | 0.01 | 0.5 | ∞ | 6 | 0.5 |
| *Ennomani et al., 2016* | 0.3–0.4 | 100 | 0.03 | 5 | 0.05 | 0 | 3.65 | 2 | 0.3 |

**Appendix 5—table 4.** Crosslinker parameters adopted in this study and in previous microstructural models that used cytosim.

| Reference | | $\rho_X$ [1/μm actin] | $K_X$ [pN/μm] | $\ell_X$ [μm] | $r_X^b$ [1/s] | $d_X^b$ [μm] | $r_X^{u,0}$ [1/s] | $f_X^u$ [pN] |
|---|---|---|---|---|---|---|---|---|
| Present work | | 15.4 | 250 | 0 | 50 | 0.02 | 50 | ∞ |
| *Cortes et al., 2020* | MM1 | 1.4–16.7 | 100 | 0.04 | 10 | 0.05 | 0.1–0.3 | 5 |
| | MM3 - MM5 | 1.7–2.8 | 100 | 0.04 | 10 | 0.05 | 0.1–0.3 | 5 |
| Cortex simulations in *Wollrab et al., 2019* | | 2–64 | 50 | 0 | 15 | 0.02 | 0.3 | 1 |
| *Bun et al., 2018* | | 2.7 | 250 | 0.01 | 10 | 0.01 | 0.1 | 3 |
| Model with turnover in *Belmonte et al., 2017* | | 0.8 | 500 | 0 | 10 | 0.01 | 0.3 | ∞ |
| *Descovich et al., 2018* | | 11.5–18.3 | 100 | 0.04 | 10 | 0.1 | 0 | 5 |
| *Ding et al., 2017* | | 0.2–11.6 | 250 | 0 | 10 | 0.01 | 0.5 | ∞ |
| *Ennomani et al., 2016* | | 0.6–1.2 | 2 | 0 | 5 | 0.03 | 0.05 | 0.05 |

