## [Editor Report · eLife Assessment]

In this study, the authors offer a theoretical explanation for the emergence of nematic bundles in the actin cortex, carrying implications for the assembly of actomyosin stress fibers. As such, the study is a **valuable** contribution to the field of actomyosin organisation in the actin cortex. The theoretical work is **solid** and provides a rigorous theoretical framework to study active self-organisation in actomyosin systems, including qualitative comparison with experimental observations.

---

## [Referee Report · Reviewer #1 (Public review)]

Summary:

In this article, Mirza et al developed a continuum active gel model of actomyosin cytoskeleton that account for nematic order and density variations in actomyosin. Using this model, they identify the requirements for the formation of dense nematic structures. In particular, they show that self-organization into nematic bundles requires both flow-induced alignment and active tension anisotropy in the system. By varying model parameters that control active tension and nematic alignment, the authors show that their model reproduces a rich variety of actomyosin structures, including tactoids, fibres, asters as well as crystalline networks. Additionally, discrete simulations are employed to calculate the activity parameters in the continuum model, providing a microscopic perspective on the conditions driving the formation of fibrillar patterns.

Strengths:

The strength of the work lies in its delineation of the parameter ranges that generate distinct types of nematic organization within actomyosin networks. The authors pinpoint the physical mechanisms behind the formation of fibrillar patterns, which may offer valuable insights into stress fiber assembly. Another strength of the work is connecting activity parameters in the continuum theory with microscopic simulations.

Weaknesses:

This paper is a very difficult read for nonspecialists, especially if you are not well-versed in continuum hydrodynamic theories. Efforts should be made to connect various elements of theory with biological mechanisms, which is mostly lacking in this paper. The comparison with experiments is predominantly qualitative. It is unclear if the theory is suited for in vitro or in vivo actomyosin systems. The justification for various model assumptions, especially concerning their applicability to actomyosin networks, requires a more thorough examination. The classification of different structures demands further justification. For example, the rationale behind categorizing structures as sarcomeric remains unclear when nematic order is perpendicular to the axis of the bands. Sarcomeres traditionally exhibit a specific ordering of actin filaments with alternating polarity patterns. Similarly, the criteria for distinguishing between contractile and extensile structures need clarification, as one would expect extensile structures to be under tension contrary to the authors' claim. Additionally, it's unclear if the model's predictions for fiber dynamics align with observations in cells, as stress fibers exhibit a high degree of dynamism and tend to coalesce with neighboring fibers during their assembly phase. Finally, it seems that the microscopic model is unable to recapitulate the density patterns predicted by the continuum theory, raising questions about the suitability of the simulation model.

---

## [Referee Report · Reviewer #2 (Public review)]

Summary:

The article by Waleed et al discusses the self-organization of actin cytoskeleton using the theory of active nematics. Linear stability analysis of the governing equations and computer simulations show that the system is unstable to density fluctuations and self-organized structures can emerge.

Strengths:

(i) Analytical calculations complemented with simulations (ii) Theory for cytoskeletal network

Weaknesses:

Not placed in the context or literature on active nematics.

Comments on revised version:

The authors have satisfactorily responded to the comments

---

## [Referee Report · Reviewer #3 (Public review)]

The manuscript "Theory of active self-organization of dense nematic structures in the actin cytoskeleton" analysis self-organized pattern formation within a two-dimensional nematic liquid crystal theory and uses microscopic simulations to test the plausibility of some of the conclusions drawn from that analysis. After performing an analytic linear stability analysis that indicates the possibility of patterning instabilities, the authors perform fully non-linear numerical simulations and identify the emergence of stripe-like patterning when anisotropic active stresses are present. Following a range of qualitative numerical observations on how parameter changes affect these patterns, the authors identify, besides isotropic and nematic stress, also active self-alignment as an important ingredient to form the observed patterns. Finally, microscopic simulations are used to test the plausibility of some of the most crucial assumptions underlying continuum simulations.

The paper is well written, figures are mostly clear, and the theoretical analysis presented in both, main text and supplement, is rigorous. Mechano-chemical coupling has emerged in recent years as a crucial element of cell cortex and tissue organization and it is plausible to think that both, isotropic and anisotropic active stresses, are present within such effectively compressible structures. Even though not explicitly stated this way by the authors, I would argue that combining these two is one of the key ingredients that distinguishes this theoretical paper from similar ones.

The diversity of patterning processes experimentally observed and theoretically described is nicely elaborated on in the introduction of the paper. The theory development and discussion of the continuum model itself is also well-embedded in a review of the relevant broad literature on active liquid crystals and active nematics, which includes plenty of previous results by the authors themselves. Interestingly, several of the patterns identified in the present work, such as 2D hexagonal and pulsatory patterns (Kumar et al, PRL, 2014), as well as contractile patches (Mietke et al, PRL 2019) have been observed previously in different, but related, active isotropic fluid models. In light of this crowded literature, the authors do good job in delineating key results obtained in the present manuscript from existing work.

The results of numerical simulations are well-presented. The discussion of numerical observations is comprehensive, but also at many times qualitative. Some of the observations resonate with recent discussions in the field, for example the observation of effectively extensile dynamics in a contractile system, which is interesting and reminiscent of ambiguities about extensile/contractile properties discussed in recent preprints (Nejad et al, Nat Comm 2024). It is convincingly concluded that, besides nematic stress on top of isotropic one, active self-alignment is a key ingredient to produce the observed patterns.

The authors must be complimented for trying to gain further mechanistic insights into their conclusions using microscopic filament simulations that were diligently performed. It is rightfully stated that these simulations only provide plausibility tests about key assumptions underlying the hydrodynamic theory. Within this scope, I would say the authors are successful. At the same time, it leaves open questions that could have been discussed more carefully. For example, I wonder what can be said about the regime \kappa>0 microscopically, in which the continuum theory does also predict the formation of stripe patterns? How does the spatial inhomogeneous organization the continuum theory predicts fit in the presented, microscopic picture and vice versa? The authors clearly explain the scope and limitations of the microscopic model, which suggests that questions like these will be interesting directions of future investigations.

Overall, the paper represents a valuable contribution to the field of active matter that should provide a fruitful basis to develop new hypothesis about the dynamic self-organisation and mechanics of dense filamentous bundles in biological systems.

---

## [Author Response]

The following is the authors’ response to the original reviews.

**eLife assessment**
In this study, the authors offer a theoretical explanation for the emergence of nematic bundles in the actin cortex, carrying implications for the assembly of actomyosin stress fibers. As such, the study is a valuable contribution to the field actomyosin organization in the actin cortex. While the theoretical work is solid, experimental evidence in support of the model assumptions remains incomplete. The presentation could be improved to enhance accessibility for readers without a strong background in hydrodynamic and nematic theories.

To address the weaknesses identified in this assessment, we have expanded the motivation and description of the theoretical model, specifically insisting on the experimental evidence supporting its rationale and assumptions. These changes in the revised manuscript are implemented in the two first paragraphs of Section “Theoretical model” and in a more detailed description and justification of the different mathematical terms that appear in that section. We have made an effort to map in our narrative different terms to mechanistic processes in the actomyosin network. Even if the nature of the manuscript is inevitably theoretical, we think that the revised manuscript will be more accessible to a broader spectrum of readers.

**Public Reviews:**

**Reviewer #1 (Public Review):**
Summary:In this article, Mirza et al developed a continuum active gel model of actomyosin cytoskeleton that account for nematic order and density variations in actomyosin. Using this model, they identify the requirements for the formation of dense nematic structures. In particular, they show that self-organization into nematic bundles requires both flow-induced alignment and active tension anisotropy in the system. By varying model parameters that control active tension and nematic alignment, the authors show that their model reproduces a rich variety of actomyosin structures, including tactoids, fibres, asters as well as crystalline networks. Additionally, discrete simulations are employed to calculate the activity parameters in the continuum model, providing a microscopic perspective on the conditions driving the formation of fibrillar patterns.Strengths:The strength of the work lies in its delineation of the parameter ranges that generate distinct types of nematic organization within actomyosin networks. The authors pinpoint the physical mechanisms behind the formation of fibrillar patterns, which may offer valuable insights into stress fiber assembly. Another strength of the work is connecting activity parameters in the continuum theory with microscopic simulations.

We thank the referee for these comments.

Weaknesses:(A) This paper is a very difficult read for nonspecialists, especially if you are not well-versed in continuum hydrodynamic theories. Efforts should be made to connect various elements of theory with biological mechanisms, which is mostly lacking in this paper. The comparison with experiments is predominantly qualitative.

We understand the point of the referee. While it is unavoidable to present the continuum hydrodynamic theory behind our results, we have made an effort in the revised manuscript to (1) motivate the essential features required from a theoretical model of the actomyosin cytoskeleton capable of describing its nematic self organization (two first paragraphs of Section “Theoretical model”), and to (2) explicitly explain the physical meaning of each of the mathematical terms in the theory, and when appropriate, relate them to molecular mechanisms in the cytoskeleton. We hope that the revised manuscript addresses the concern of the referee.

Regarding the comparison with experiments, they are indeed qualitative because the main point of the paper is to establish a physical basis for the self-organization of dense nematic structures in actomyosin gels. Somewhat surprisingly, we argue that a compelling mechanism explaining the tendency of actomyosin gels to form patterns of dense nematic bundles has been lacking. As we review in the introduction, these patterns are qualitatively diverse across cell types and organisms in terms of geometry and dynamics, and for this reason, our goal is to show that the same material in different parameter regimes can exhibit such qualitative diversity. A quantitative comparison is difficult for several reasons. First, many of the parameters in our theory have not been measured and are expected to vary wildly between cell types. In fact, estimates in the literature often rely on comparison with hydrodynamic models such as ours. For this reason, we chose to delineate regimes leading to qualitatively different emerging architectures and dynamics. Second, the patterns of nematic bundles found across cell types depend on the interaction between (1) the intrinsic tendency of actomyosin gels to form such structures studied here and (2) other elements of the cellular context. For instance, polymerization and retrograde flow from the lamellipodium, the physical barrier of the nucleus, and the interaction with the focal adhesion machinery are essential to understand the emergence of stress fibers in adherent cells. Cell shape and curvature anisotropy control the orientation of actin bundles in parallel patterns in the wings and trachea of insects. Nuclear positions guide the actin bundles organizing the cellularization of Sphaeroforma arctica [11]. Here, we focus on establishing that actomyosin gels have an intrinsic ability to self organize into dense nematic bundles, and leave how this property enables the morphogenesis of specific structures for future work. We have emphasized this point in the revised section of conclusions.

(B) It is unclear if the theory is suited for in vitro or in vivo actomyosin systems. The justification for various model assumptions, especially concerning their applicability to actomyosin networks, requires a more thorough examination.

We thank the referee for this comment. Our theory is applicable to actomyosin gels originating from living cells. To our knowledge, the ability of reconstituted actomyosin gels from purified proteins to sustain the kind of contractile dynamical steady-states observed in living cells is very limited. In the revised manuscript, we cite a very recent preprint presenting very exciting but partial results in this direction [49]. Instead, reconstituted in vitro systems encapsulating actomyosin cell extracts robustly recapitulate contractile steady-states. This point has been clarified in the first paragraph of Section “Theoretical model”.

(C) The classification of different structures demands further justification. For example, the rationale behind categorizing structures as sarcomeric remains unclear when nematic order is perpendicular to the axis of the bands. Sarcomeres traditionally exhibit a specific ordering of actin filaments with alternating polarity patterns.

We agree with the referee and in the revised manuscript we have avoided the term “sarcomeric” because it refers to very specific organizations in cells. What we previously called “sarcomeric patterns”, where bands of high density exhibit nematic order perpendicular to the axis of the bands, is not a structure observed to our knowledge in cells. It is introduced to delimit the relevant region in parameter space. In the revised manuscript, we refer to this pattern as “banded pattern with perpendicular nematic organization” or “banded pattern” in short.

(D) Similarly, the criteria for distinguishing between contractile and extensile structures need clarification, as one would expect extensile structures to be under tension contrary to the authors' claim.

We thank the referee for raising this point, which was not sufficiently clarified in the original manuscript. We first note that in incompressible active nematic models, active tension is deviatoric (traceless and anisotropic) because an isotropic component would simply get absorbed by the pressure field enforcing incompressibility. Being compressible, our model admits an active tension tensor with deviatoric and isotropic components. We consider always a contractile (positive) isotropic component of active tension, but the deviatoric component can be either contractile (𝜅 > 0) or extensile (𝜅 < 0), where we follow the common terminology according to which in contractile/extensile active nematics the active stress is proportional to q with a positive/negative proportionality constant [see e.g. https://doi.org/10.1038/s41467018-05666-8]. Furthermore, as clarified in the revised manuscript, total active stresses accounting for the deviatoric and isotropic components are always contractile (positive) in all directions, as enforced by the condition |𝜅| < 1.

For fibrillar patterns, we need 𝜅 < 0, and therefore active stresses are larger perpendicular to the nematic direction. This means that the anisotropic component of the active tension is extensile, although, accounting for the isotropic component, total active tension is contractile (see Fig. 1c). This is now clarified in the text following Eq. 7 and in Fig. 1.

However, following fibrillar pattern formation and as a result of the interplay between active and viscous stresses, the total stress can be larger along the emergent dense nematic structures (“contractile structures”) or perpendicular to them (“extensile structures”). To clarify this point, in the revised Fig. 4 and the text referring to it, we have expanded our explanation and plotted the difference between the total stress component parallel to the nematic direction (𝜎∥) and the component perpendicular to the nematic direction (𝜎⊥), with contractile structures satisfying 𝜎∥ − 𝜎⊥ > 0 and extensile structures satisfying 𝜎∥ − 𝜎⊥ < 0. See lines 280 to 303. This is consistent with the common notion of contractile/extensile systems in incompressible nematic systems [see e.g. https://doi.org/10.1038/s41467-018-05666-8].

(E) Additionally, its unclear if the model's predictions for fiber dynamics align with observations in cells, as stress fibers exhibit a high degree of dynamism and tend to coalesce with neighboring fibers during their assembly phase.

In the present work, we focus on the self-organization of a periodic patch of actomyosin gel. However, in adherent cells boundary conditions play an essential role, as discussed in our response to comment (A) by this referee. In ongoing work, we are studying with the present model the dynamics of assembly and reconfiguration of dense nematic structures in domains with boundary conditions mimicking in adherent cells, possibly interacting with the adhesion machinery, finding dynamical interactions as those suggested by the referee. As an example, we show a video of a simulation where at the edge of the circular domain, there is an actin influx modeling the lamellipodium, and in four small regions friction is higher simulating focal adhesions. Under these boundary conditions, the model presented in the paper exhibits the kind of dynamical reorganizations alluded by the referee.

**Author response video 1.**

We would like to note, however, that the prominent stress fibers in cells adhered to stiff substrates, so abundantly reported in the literature, are not the only instance of dense nematic actin bundles. In the present manuscript, we emphasize the relation of the predicted organizations with those found in different in vivo contexts not related to stress fibers, such as the aligned patterns of bundles in insects (trachea, scales in butterfly wings), in hydra, or in reproductive organs of C elegans; the highly dynamical network of bundles observed in C elegans early embryos; or the labyrinth patters of micro-ridges in the apical surface of epidermal cells in fish.

(F) Finally, it seems that the microscopic model is unable to recapitulate the density patterns predicted by the continuum theory, raising questions about the suitability of the simulation model.

We thank the referee for raising this question, which needs further clarification. The goal of the microscopic model is not to reproduce the self-organized patterns predicted by the active gel theory. The microscopic model lacks essential ingredients, notably a realistic description of hydrodynamics and turnover. Our goal with the agent-based simulations is to extract the relation between nematic order and active stresses for asmall homogeneous sample of the network. This small domain is meant to represent the homogeneous active gel prior to pattern formation, and it allows us to substantiate key assumptions of the continuum model leading to pattern formation, notably the dependence of isotropic and deviatoric components of the active stress on density and nematic order (Eq. 7) and the active generalized stress promoting ordering.

We should mention that reproducing the range of out-of-equilibrium mesoscale architectures predicted by our active gel model with agent-based simulations seems at present not possible, or at least significantly beyond the state-of-the-art. To our knowledge, these models have not been able to reproduce the heterogeneous nonequilibrium contractile states involving sustained self-reinforcing flows underlying the pattern formation mechanism studied in our work. The scope of the discrete network simulations has been clarified in lines 340 to 349 in the revised manuscript.

While agent-based cytoskeletal simulations are very attractive because they directly connect with molecular mechanisms, active gel continuum models are better suited to describe out-of-equilibrium emergent hydrodynamics at a mesoscale. We believe that these two complementary modeling frameworks are rather disconnected in the literature, and for this reason, we have attempted substantiate some aspects of our continuum modeling with discrete simulations. We have emphasized the complementarity of the two approaches in the conclusions.

**Reviewer #1 (Recommendations For The Authors):**
Questions on the theory:Does rho describe the density of actin or myosin? The authors say that they are modeling actomyosin material as a whole, but the actin and myosin should be modeled separately. Along, similar lines, does Q define the ordering of actin or myosin?

Active gel models of the actomyosin cytoskeleton have been formulated with independent densities for actin and for myosin or using a single density field, implicitly assuming a fixed stoichiometry. Super-resolution imaging of the actomyosin cytoskeleton also suggest that in principle it makes sense to consider different nematic fields for actin and for myosin filaments. In the revised manuscript, we now explicitly mention that our density and nematic field are effective descriptions of the entire actomyosin gel (lines 82-84).

A more detailed model would entail additional material parameters, not available experimentally, which may help reproduce specific experiments but that would make the systematic study of the different behaviors much more difficult. Our approach has been to keep the model minimal meeting the fundamental requirements outlined in the first paragraphs of Section “Theoretical model”.

Should the active stress depend on material density? It seems strange (from Eq. 3) that active stress could be non-zero even where density is zero, since sigma_act does not depend on rho.

Yes, active stress is assumed to be proportional to density. Eq. 3 in the original manuscript was misleading (it was multiplied by rho in Eq. 2). In the revised manuscript, we have explained with a bit more detail the theoretical model, clarifying this point.

The authors should clearly explain their rationale for retaining certain types of nonlinear terms while ignoring others in theory. For instance, the nonlinearities in the equations of motion are sometimes quadratic in the fields, while there are also some cubic terms. Please remark up to what order in the fields the various interactions are modeled.

We thank the referee for raising this point. The nonlinearities in the theory are easily explained on the basis of a small number of choices. We have added a new paragraph towards the end of Section “Theoretical model” (lines 145 to 152) providing a rationale for the origin and underlying assumptions leading to different nonlinearities.

To connect with experiments and the biological context, please explain the biological origin of various terms in the model: (1) L-dependent terms in Eq. 2 and 4, (2) Flowalignment of nematic order and experimental evidence in support of it, (3) densitydependent susceptibility terms in Eq. 4

(1) Unfortunately, the L-dependent terms are very bulky, but are very standard in nematic theories. The best way to understand their physical significance is through the expression of the nematic free-energy, which is now given and explained in the revised manuscript (Eq. 3). The resulting complicated expression for the molecular field and the nematic stress (Eqs. 4 and 5) are mathematical consequences of the choice of nematic free energy. In the revised manuscript, we also attempt to provide a basis for these terms in the context of the actin cytoskeleton. (2) To our knowledge, the best reference supporting this term from experiments is Reymann et al, eLife (2016). In the revised manuscript, we have provided a physical interpretation. (3) We have expanded the motivation and plausible microscopic justification of this term.

There are different 'activity' terms in the model. Their biophysical origin is not made clear. For example, the authors should make clear if these activities arise from filament or motor activity. Relatedly, the authors should provide a comprehensive discussion of the signs of the different active parameters and their physical interpretations.

In an active gel model, activity parameters are phenomenological and how they map to molecular mechanisms is not precisely known, although conventionally contractile active tension is ascribed to the mechanical transduction of chemical power by myosin motors. The fact is that, besides myosin activity, there are many nonequilibrium processes in the actomyosin cytoskeleton that may lead to active stresses including (de)polymerization of filaments or (un)binding of crosslinkers. In the revised manuscript, we have added sentences illustrating how different terms may result from microscopic mechanisms, but providing a precise mapping between our model and nonequilibrium dynamics of proteins is beyond the scope of our work, although our discrete network simulations address this issue to a certain degree.

Following the suggestion of the referee, our description of the theory now discusses much more extensively the signs of activity parameters and their physical interpretations, e.g. the text following Eq. 7.

Throughout the paper, various activity terms are varied independently of each other. Is that a reasonable assumption given that activities should depend on ATP and are thus not independent of one another?

We agree that, ultimately, all active process depend on the conversion of chemical energy into mechanical energy. However, recent work has highlighted how active tension also depends on the microscopic architecture of the network controlled by multiple regulators of the actomyosin cytoskeleton (e.g. Chug et al, Nat Cell Biol, 2017). It is reasonable to expect that, for a given rate of ATP consumption, chemical power will be converted into mechanical power in different ways depending on the micro-architecture of the cytoskeleton, e.g. the stoichiometry of filaments, crosslinkers, myosins, or the length distribution of filaments (very long filaments crosslinked by myosins may be difficult to reorient but may contract efficiently).

We have added a paragraph in Section “Theoretical model” with a discussion, lines 153 to 156.

Sarcomeres are muscle fibers that exhibit alternating polarity pattern. Such patterning is not evident in what the authors call 'sarcomeres' in Fig. 2. I believe the authors should revise their terminology and not loosely interpret existing classifications in the field.

We thank the referee for raising this point. We have changed the terminology.

Fig 2a: Is the cartoon for filament alignment incorrect for kappa>0?

The cartoon is correct. In the revised manuscript we have explained more clearly the physical meaning of kappa in the text following Eq. 7. In the caption of Fig. 1 and of Fig. 2a, we have also clarified that when the absolute value of kappa is <1, then active tension is positive in all directions.

Within the section "Requirements for fibrillar and banded patterns", it will be useful to show the figures for varying the different active parameters in the main figures.

We have followed the referee’s suggestion and moved Supp. Fig. 1 of the original manuscript to the main figures.

How do the authors decide if bundles are contractile or extensile? Why are contractile bundles under tension while extensile bundles are under compression? I would expect the opposite.

We agree that this point deserves a more detailed explanation. In the revised manuscript and in the new Figure 4, we further develop this point. The fibrillar pattern forms when kappa<0. We further assume that -1<kappa<0, so that active tension is positive in all directions. In this regime, the deviatoric (anisotropic) part of active tension is extensile. However, following pattern formation and because of the interplay between active and viscous stresses, the total stress in the emerging bundles may become extensile or contractile, depending on whether the largest component of stress is perpendicular or along the bundle axis. This is now presented in the updated figure, with new panels presenting maps of the total tension. The text discussing this point has been rewritten and we hope that the new version is much clearer (lines 280 to 303).

A contractile bundle tends to shorten, but it cannot do it because of boundary conditions or the interaction with other bundles. As a result they are in tension. Conversely, an extensile bundle tries to elongate, but being constrained, it becomes compressed. As an analogy, consider the cortex of a suspended cell. The cortex is contractile, but it cannot contract because of volume regulation in th cell, which is typically pressurized. As a result, tension in the cortex is positive, as shown by Laplace’s law [10.1016/j.tcb.2020.03.005]. We have tried to clarify this point in the revised manuscript.

Can the authors reproduce alternating density patterns using the cytosim simulations? This is an important step in establishing the correspondence between the continuum theory and the agent-based model.

We have addressed this point in our response to public comment (F) of this referee.

The authors do not provide code or data.

The finite element code with an input file require to run a representative simulation in the paper is now made available, see Ref. [74].

The customizations of Cytosim needed to account for nematic order in our discrete network simulations are available, see Ref. [98].

**Reviewer #2 (Public Review):**
Summary:The article by Waleed et al discusses the self organization of actin cytoskeleton using the theory of active nematics. Linear stability analysis of the governing equations and computer simulations show that the system is unstable to density fluctuations and self organized structures can emerge. While the context is interesting, I am not sure whether the physics is new. Hence I have reservations about recommending this article.

We thank the referee for these comments. In the revised manuscript, we have highlighted the novelty, particularly in the last paragraph of the introduction, the first two paragraphs of Section “Theoretical model”, and in the conclusions. Despite a very large literature on theoretical models of stress fibers, actin rings, and active nematics, we argue that the active self-organization of dense nematic structures from an isotropic and low-density gel has not been compellingly explained so far. Many models assume from the outset the presence of actin bundles, or explain their formation using localized activity gradients. The literature of active nematics has extensively studied symmetry breaking and the self-organization. However, most of the works assume initial orientational order. Only a few works study the emergence of nematic order from a uniform isotropic state, but consider dry systems lacking hydrodynamic interactions or incompressible and density-independent systems [37,38]. Yet, pattern formation in actomyosin gels is characterized by large density variations, and by highly compressible flows, which coordinate in a mechanism relying on an advective instability and self-reinforcing flows.

Our theoretical model is not particularly novel, and as we mention in the manuscript, it can be particularized to different models used in the literature. However, we argue that it has the right minimal features to capture nematic self-organization in actomyosin gels. To our knowledge, no previous study explains the emergence of dense and nematic structures from a low-density isotropic gel as a result of activity and involving the advective instability typical of symmetry-breaking and patterning in the actomyosin cytoskeleton. These are important qualitative features of our results that resonate with a large experimental record, and as such, we believe that our work provides a new and compelling mechanism relying on self-organization to explain the prominence and diversity of patterns involving dense nematic bundles in the actomyosin cytoskeleton across species.

Strengths:(i) Analytical calculations complemented with simulations (ii) Theory for cytoskeletal networkWeaknesses:Not placed in the context or literature on active nematics.

We agree with the referee that this was a weakness of the original manuscript. In the revised manuscript, within reasonable space constraints given the size and dynamism of the field of active nematics, we have placed our work in the context of this field (end of introduction and first two paragraphs of Section “Theoretical model”). The published version of our companion manuscript [45] also contributes to providing a clear context to our theoretical model within the field.

**Reviewer #2 (Recommendations For The Authors):**
The article by Waleed et al discusses the self organization of actin cytoskeleton using the theory of active nematics. Linear stability analysis of the governing equations and computer simulations show that the system is unstable to density fluctuations and self organized structures can emerge. While the context is interesting, I am not sure whether the physics is new. Hence I have reservations about recommending this article. I explain my questions comments below.

We have responded to this comment above.

(i) Active nematics including density variations have been dealt quite extensively in the literature. For example, the works of Sriram Ramaswami have dealt with this system including linear stability analysis, simulations etc. In what way is the present work different from the system that they have considered?(ii) Active flows leading to self organization has been a topic of discussion in many works. For example: (i) Annual Review of Fluid Mechanics, Vol. 43:637-659, 2010, https://doi.org/10.1146/annurev-fluid-121108-145434 (ii) S Santhosh, MR Nejad, A Doostmohammadi, JM Yeomans, SP Thampi, Journal of Statistical Physics 180, 699-709 (iii) M. G. Giordano1, F. Bonelli2, L. N. Carenza1,3, G. Gonnella1 and G. Negro1, Europhysics Letters, Volume 133, Number 5. In what way this work is different from any of these?(iii) I am confused about the models used in the paper. There is significant literature from Prof. Mike Cates group, Prof. Julia Yeomans group, Prof. Marchetti's group who all use similar governing equations. In the present paper, I find it hard to understand whether the model used is similar to the existing ones in literature or are there significant differences. It should be clarified.

Response to (i), (ii) and (iii).

We completely agree with this referee (and also the previous referee), that the contextualization of our work in the field of active nematics was very insufficient. In the revised manuscript, the last paragraph of the introduction and the first two paragraphs of Section “Theoretical model” now address this point. In short, previous active nematic models predicting patterns with density variations have been either for dry active matter (disregarding hydrodynamic interactions), or for suspensions of active particles moving in an incompressible flow. None of these previous works predict nematic pattern formation as a result of activity relying on the advective instability and self-reinforcing compressible flows, leading to high density and high order bundles surrounded by an isotropic low density phase. Yet, these are fundamental features observed in actomyosin gels. Many works deal with symmetry-breaking of a system with pre-existing order, but very few address how order emerges actively from an isotropic state. We thank the referee for pointing at the paper by Santhosh et al, who nicely make this argument and is now cited. Our mechanism is fundamentally different from that in Santhosh, whose model is incompressible and ignores density variations.

We hope that the revised manuscript addresses this important concern.

(i) >(iv) Below Eqn 6, it starts by saying that the “...origin..is clear...” Its not. I don't understand the physical origin of the instability, and this should be clarified, may be with some illustrations.

We apologize for this unfortunate sentence, which we have rewritten in the revised manuscript (lines 181 to 185).

**Reviewer #3 (Public Review):**
The manuscript "Theory of active self-organization of dense nematic structures in the actin cytoskeleton" analysis self-organized pattern formation within a two-dimensional nematic liquid crystal theory and uses microscopic simulations to test the plausibility of some of the conclusions drawn from that analysis. After performing an analytic linear stability analysis that indicates the possibility of patterning instabilities, the authors perform fully non-linear numerical simulations and identify the emergence of stripelike patterning when anisotropic active stresses are present. Following a range of qualitative numerical observations on how parameter changes affect these patterns, the authors identify, besides isotropic and nematic stress, also active self-alignment as an important ingredient to form the observed patterns. Finally, microscopic simulations are used to test the plausibility of some of the conclusions drawn from continuum simulations.The paper is well written, figures are mostly clear and the theoretical analysis presented in both, main text and supplement, is rigorous. Mechano-chemical coupling has emerged in recent years as a crucial element of cell cortex and tissue organization and it is plausible to think that both, isotropic and anisotropic active stresses, are present within such effectively compressible structures. Even though not yet stated this way by the authors, I would argue that combining these two is of the key ingredients that distinguishes this theoretical paper from similar ones. The diversity of patterning processes experimentally observed is nicely elaborated on in the introduction of the paper, though other closely related previous work could also have been included in these references (see below for examples).

We thank the referee for these comments and for the suggestion to emphasize the interplay of isotropic and anisotropic active tension, which is possible only in a compressible gel, as mentioned in the revised manuscript. We have emphasized this point in different places in the revised manuscript. We thank the suggestions of the referee to better connect with existing literature.

To introduce the continuum model, the authors exclusively cite their own, unpublished pre-print, even though the final equations take the same form as previously derived and used by other groups working in the field of active hydrodynamics (a certainly incomplete list: Marenduzzo et al (PRL, 2007), Salbreux et al (PRL, 2009, cited elsewhere in the paper), Jülicher et al (Rep Prog Phys, 2018), Giomi (PRX, 2015),...). To make better contact with the broad active liquid crystal community and to delineate the present work more compellingly from existing results, it would be helpful to include a more comprehensive discussion of the background of the existing theoretical understanding on active nematics. In fact, I found it often agrees nicely with the observations made in the present work, an opportunity to consolidate the results that is sometimes currently missed out on. For example, it is known that self-organised active isotropic fluids form in 2D hexagonal and pulsatory patterns (Kumar et al, PRL, 2014), as well as contractile patches (Mietke et al, PRL 2019), just as shown and discussed in Fig. 2. It is also known that extensile nematics, \kappa<0 here, draw in material laterally of the nematic axis and expel it along the nematic axis (the other way around for \kappa>0, see e.g. Doostmohammadi et al, Nat Comm, 2018 "Active Nematics" for a review that makes this point), consistent with all relative nematic director/flow orientations shown in Figs. 2 and 3 of the present work.

We thank the referee for these suggestions. Indeed, in the original submission we had outsourced much of the justification of the model and the relevant literature to a related pre-print, but this is not reasonable. The companion publication has now been accepted in the New Journal of Physics, with significant changes to better connect the work to the field of active nematics. A preprint reflecting those changes is available in Ref. [64], but we hope to reference the published paper that will come out soon.

In the revised manuscript, we have significantly rewritten the Section “Theoretical model” to frame the continuum model in the context of the field of active nematics. While our model and results have commonalities with previous work, there are also important differences. We have highlighted the novelty of the present work along with the relation with previous studies and theoretical models in the last paragraph of the introduction and the first two paragraphs of Section “Theoretical model”. Furthermore, as suggested by the referee, we have made an effort to connect our results with previous work by Kumar, Mietke, Doostmohammadi and others.

Regarding the last point alluded by the referee (“extensile nematics, \kappa<0 here, draw in material laterally of the nematic axis and expel it along the nematic axis”), the picture raised by the referee would be nuanced for our compressible system as compared to the incompressible systems discussed in that reference. As we have elaborated in our response to point (D) of Referee #1, our systems are overall contractile (with positive active tension in all directions), but the deviatoric component of the active tension can be either extensile or contractile. In our “extensile” models (left in Fig. 2c), material is drawn to laterally to the nematic axis but it is not expelled along this axis. Instead, it is “expelled” by turnover. In the revised manuscript, we have added a comment about this.

The results of numerical simulations are well-presented. Large parts of the discussion of numerical observations - specifically around Fig. 3 - are qualitative and it is not clear why the analysis is restricted to \kappa<0. Some of the observations resonate with recent discussions in the field, for example the observation of effectively extensile dynamics in a contractile system is interesting and reminiscent of ambiguities about extensile/contractile properties discussed in recent preprints (https://arxiv.org/abs/2309.04224). It is convincingly concluded that, besides nematic stress on top of isotropic one, active self-alignment is a key ingredient to produce the observed patterns.

We thank the referee for these comments. We are reluctant to extend the detailed analysis of emergent architectures and dynamics to the case \kappa > 0 as it leads to architectures not observed, to our knowledge, in actin networks. In the revised manuscript, we have expanded and clarified the characterization of emergent contractile/extensile networks by reporting the relative magnitude of stress along and perpendicular to the nematic direction. Our revised manuscript clearly shows that even though all of our simulations describe locally contractile systems with extensile anisotropic active tension, the emergent meso-structures can be either extensile or contractile, with the extensile ones exhibiting the usual bend-type instability (a secondary instability in our system) described classically for extensile active nematic systems. We have rewritten the text discussing this (lines 280 to 303), where we have placed these results in the context of recent work reporting the nontrivial relation between the contractility/extensibility of the local units vs the nematic pattern.

I compliment the authors for trying to gain further mechanistic insights into this conclusion with microscopic filament simulations that are diligently performed. It is rightfully stated that these simulations only provide plausibility tests and, within this scope, I would say the authors are successful. At the same time, it leaves open questions that could have been discussed more carefully. For example, I wonder what can be said about the regime \kappa>0 (which is dropped ad-hoc from Fig. 3 onward) microscopically, in which the continuum theory does also predict the formation of stripe patterns - besides the short comment at the very end? How does the spatial inhomogeneous organization the continuum theory predicts fit in the presented, microscopic picture and vice versa?

We thank the referee for this compliment. We think that the point raised by the referee is very interesting. It is reasonable to expect that the sign of \kappa may not be a constant but rather depend on S and \rho. Indeed, for a sparse network with low order, the progressive bundling by crosslinkers acting on nearby filaments is likely to produce a large active stress perpendicular to the nematic direction, whereas in a dense and highly ordered region, myosin motors are more likely to effectively contract along the nematic direction whereas there is little room for additional lateral contraction by additional bundling. As discussed in our response to referee #1, we believe that studying the formation of patterns using the discrete network simulations is far beyond the scope of our work. We discuss in lines 332 to 341, as well as in the last paragraph of the conclusions, the scope and limitations of our discrete network simulations.

Overall, the paper represents a valuable contribution to the field of active matter and, if strengthened further, might provide a fruitful basis to develop new hypothesis about the dynamic self-organisation of dense filamentous bundles in biological systems.
**Reviewer #3 (Recommendations For The Authors):**
The statement "the porous actin cytoskeleton is not a nematic liquid-crystal because it can adopt extended isotropic/low-order phases" is difficult to understand and should be clarified, as the next paragraph starts formulating a nematic active liquid crystal theory. Do the authors mean a crystal that "Tends to be in a disordered phase?", according to its equilibrium properties? It would still be a "nematic liquid crystal", only its ground state is not a nematic phase.

We agree with the referee, and we hope that changes in the introduction and in Section“Theoretical model” address this comment.

I could not find what Frank energy is precisely used, that would be helpful information.

In the revised manuscript, we have provided the expression for the nematic free energy in Eq. 3.

The Significance of green/purple arrows in Fig 2a sketch unclear, green arrows also in b,c, do they represent the same quantity? From the simulations images it is overall it is very difficult to see how the flows are oriented near the high-density regions (i.e. if they are towards / away from the strip).

We thank the referee for bringing this up. The colorcodings of the sketches were confusing. The modified figures (Fig. 1(c) and Fig. 2(a)) present now a clearer and unified representation of anisotropic tension. The green arrows in Fig. 2(c) represent the out-of-equilibrium flows in the steady state. We agree that the zoom is insufficient to resolve the flow structure. For this reason, in the revised Fig. 2, we have added additional panels showing the flow with higher resolution.

It is currently unclear how the linear stability results - beyond identification of the parameter \delta - inform any of the remaining manuscript. Quantitative comparisons of the various length scales seen in simulated patterns (e.g. Fig. 2b, 3c etc) with linear predictions and known characteristic length scales would be instructive mechanistically, would make the overall presentation more compelling and probes limitations of linear results.

In the revised manuscript, we have provided further information so that the readers can appreciate the predictions and limitations of the linear stability results. We have added a sentence and a Figure to show that, in addition to the critical activity, the linear theory provides a good prediction of the wavelengh of the pattern. See lines 199 to 201.

It is not clear what is meant by "[bundle-formation] requires that active tension perpendicular to nematic orientation is larger than along this direction", and therefore also not why that would be "counter-intuitive". If interpreted naively, I would say that a large tension brings in more filaments into the bundle, so that may well be an obviously helpful feature for bundle formation and maintenance. In any case, it would be helpful if clarity is improved throughout when arguments about "directions of tensions" are made.

We have significantly rewritten the first paragraphs of section “Microscopic origin…” to clarify this point (lines 330 to 339). This paragraph, along with other changes in the manuscript such as the explanation of Eq. 7 or the discussion about the stress anisotropy in the new version of Fig. 4 (see lines 280 to 303), provide a better explanation of this important point.

All density color bars: Shouldn't they rather be labelled \rho/\rho_0?

Yes! We have corrected this typo.

Scalar product missing in caption definition of order parameter Fig. 2

We have corrected this typo.

Fig. 3a: I suggest to put the expression for q0 in the caption

We have changed q_0 by S_0 and clarified its meaning in the caption of what now is Fig 4.

Paragraph on bottom right of page 6 should several times probably refer to Fig.3c(...), instead of Fig. 3b

We have corrected this typo./